# PM2.5 surface concentrations in southern West African urban areas based on sun photometer and satellite observations

Jean-François Léon[1], Aristide Barthélémy Akpo[2], Mouhamadou Bedou[3], Julien Djossou[2], Marleine Bodjrenou[2], Véronique Yoboué[3], and Cathy Liousse[1]

[1]Laboratoire d'Aérologie, Université Paul Sabatier, CNRS, Toulouse, France
[2]Laboratoire de Physique du Rayonnement, Université d'Abomey Calavi, BP 526, Cotonou, Bénin
[3]Laboratoire de Physique de l'atmosphère, Université Félix-Houphouët-Boigny, Abidjan, Côte d'Ivoire

**Correspondence:** Jean-François Léon (jean-francois.leon@aero.obs-mip.fr)

**Abstract.** Southern West Africa (sWA) is influenced by large amounts of aerosol particles of both anthropogenic and natural origins. Anthropogenic aerosol emissions are expected to increase in the future due to the economical growth of African megacities. In this paper, we investigate the aerosol optical depth (AOD) in the coastal area of the Gulf of Guinea using sun photometer and MODIS satellite observations. A network of lightweight handheld sun photometers have been deployed in sWA from December 2014 to April 2017 at 5 different locations in Côte d'Ivoire and Bénin. The handheld sun photometer measures the solar irradiance at 465, 540 and 619 nm and is operated manually once per day. Handheld sun photometer observations are complemented by available AERONET sun photometer observations and MODIS level 3 time series between 2003 and 2019. MODIS daily level 3 AOD agrees well with sun photometer observations in Abidjan and Cotonou (correlation coefficient R=0.89 and RMSE=0.19). A classification based on the sun photometer AOD and Ångström exponent (AE) is used to separate the influence of coarse mineral dust and urban-like aerosols. The AOD seasonal pattern is similar for all the sites and is clearly influenced by the mineral dust advection from December to May. Sun photometer AODs are analyzed in coincidence with surface PM2.5 concentrations to infer trends in the particulate pollution levels over conurbation of Abidjan (Côte d'Ivoire) and Cotonou (Bénin). PM2.5 to AOD conversion factors are evaluated as a function of the season and the aerosol type identified in the AE classification. Highest PM2.5 concentrations (up to 300 $\mu g/m^3$) are associated to the advection of mineral dust in the heart of the dry season (December-February). Annual means are around 30 $\mu g/m^3$ and 80% of days in the winter dry season have a value above 35 $\mu g/m^3$ while concentrations remains below 16 $\mu g/m^3$ from May to September. No obvious trend is observed in the 2003-2019 MODIS-derived PM2.5 time series. However the short dry period (August-September) when urban-like aerosols dominate, is associated to a monotonic trend between 0.04 and 0.43 $\mu g/m^3/year$ in the PM2.5 concentrations over the period 2003-2017. The monotonic trend remains uncertain but is coherent with the expected increase in combustion aerosol emissions in sWA.

*Copyright statement.* TEXT

# 1 Introduction

The increasing trend in the anthropogenic emissions in Africa (Liousse et al., 2014) gives rise to the question of the impact of human activities on air quality, the monsoon system and the regional climate. The Gulf of Guinea and adjacent countries, hereinafter called southern West Africa (sWA), is influenced by large amounts of aerosol particles of both anthropogenic and natural origins advected from the African continent. The season cycle in sWA is driven by the monsoon system (Knippertz et al., 2015) with the alternation of a major winter (November to March) dry season and a summer (June-July) rainy season. The Inter Tropical Front (Lélé and Lamb, 2010) is at its southernmost position during the winter dry season enabling the northeasterly Harmattan wind to carry a dust-laden dry air southward (Adetunji et al., 1979). The major conurbations of sWA are then downwind the mineral dust emission of the Bodélé depression in Chad, the predominant dust emission source of West Africa (Todd et al., 2007; Washington, 2005; Koren et al., 2006; Schepanski et al., 2009). Carbonaceous aerosols that are emitted by open biomass burning (Liousse et al., 2010) are also advected southward to the main coastal cities of sWA during the dry period. The summer wet season corresponds to the continental intrusion of the southwesterly monsoon winds carrying moist air and precipitation. During this period, biomass burning emissions in central Africa can be advected to sWA by easterly wind and thus can impact the local air quality of coastal conurbations (Menut et al., 2018).

sWA is a hot spot of atmospheric aerosol concentrations as revealed by satellite-derived aerosol optical depth (Kaufman et al., 2002; Mehta et al., 2018). Atmospheric aerosols can alter the development of monsoons by weakening the land-ocean thermal contrast, and the thermodynamic stability and the convective potential of the lower atmosphere (Li et al., 2016). Precipitation reduction in the West African monsoon region (Janicot, 1992) has been attributed to high aerosol concentrations near the Gulf of Guinea (Huang et al., 2009). Yoon et al. (2016) have pointed out the role of carbonaceous aerosols on rainfall reduction in the West African monsoon region. Aerosol effects on regional climate falls into two categories (Boucher et al., 2013). Direct effects refer to the influence of aerosol scattering and absorption on the atmospheric radiative balance. Indirect and semi-direct effects refer to the impact of aerosol on cloud properties with subsequent effects on the radiative balance. The aerosol optical depth (AOD) is one of the key parameters for assessing the aerosol direct radiative impact (Liou, 2002).

AOD is the primary aerosol optical parameter derived from satellite remote sensing (Kaufman et al., 1997). AOD is related to the reduction in the atmospheric transmission due to aerosol particles in suspension in the atmosphere. AOD can be measured directly from the ground by using a sun photometer (Volz, 1959; Prospero et al., 1979; Tanré et al., 1988; Nakajima et al., 1996). The Aerosol Robotic Network (Holben et al., 1998, 2001) is one of the most important federated network of ground-based automatic sun photometers providing continuous AOD measurements in many places of the world. Western Africa benefits from a good geographical coverage of AERONET sun photometers in the Sahel transect. The stations are located in remote places dedicated to the monitoring of Saharan dust or biomass burning aerosols optical properties and atmospheric transport (Tanré et al., 1988; Redelsperger et al., 2006; Mallet et al., 2008; Léon et al., 2009). However sun photometer observations in the large conurbations surrounding the gulf of Guinea remain scarce. AOD observations in the coastal part of sWA will thus provide additional ground-truths for satellite validation.

Long-term satellite-derived AOD can also make up for the lack of in situ particulate matter (PM) surface observations. As air quality in sWA conurbations is still poorly covered by operational observational networks, satellite-derived PM may have a significant added-value for air quality monitoring. There is an abundant literature on linking columnar satellite AOD to PM (Kacenelenbogen et al., 2006; Hoff and Christopher, 2009; Ma et al., 2015; van Donkelaar et al., 2016). The relationship between instantaneous AOD and PM measurements is not straightforward and several regression models have been tested, either linear (Kacenelenbogen et al., 2006), multi-linear (Gupta and Christopher, 2009) or non-linear (Gupta and Christopher, 2009; Yahi et al., 2013; Kamarul Zaman et al., 2017). The conversion model from AOD to PM depends on the aerosol physical properties (aerosol type), hygroscopicity and the atmospheric dynamics including boundary layer mixing. In particular, the variability of the planeraty boundary layer depth can act as a controlling factor to the ratio between surface PM and columnar AOD (Boyouk et al., 2010; Sayer et al., 2016). Vertical profil of aerosols and meteorological parameters affects the correlation between PM and AOD (Sinha et al., 2015). Additional local analysis of the PM-to-AOD relationship based on in situ observations will strengthen the systematic retrieval of PM for satellite remote sensing.

In a companion paper (Djossou et al., 2018), the AOD measurements obtained downtown major cities of Abidjan (Côte d'Ivoire) and Cotonou (Bénin) were presented along with the surface observations of the PM2.5 mass concentration and carbonaceous aerosol composition. A tentative analysis of the relationship between AOD and PM2.5 was made and show the potential of AOD to infer PM2.5 concentration in both conurbations. In this paper, we report additional AOD measurements over sWA using lightweight handheld and automatic sun photometer with the purpose of validating the MODIS-derived AOD at the regional scale and investigating further the use of AOD for local pollution assessment. Section 2 presents the data sets and the methods. Section 3 presents the sun photometer time series and the validation of the satellite AODs. The relationship between AOD and PM2.5 is investigated in section 4. The last section presents the interannual trends in PM2.5 derived from the MODIS observations.

## 2 Data and Method

All the observations were acquired in a geographical box ranging from approximately $4°$ N to $9°$ N and $6°$ W to $5°$ E (Figure 1). SWA has a marked latitudinal gradient in ecosystems that largely impacts the emission and deposition of particles and trace gases (Adon et al., 2010). We define SWA as delimited by the shore of gulf of Guinea and $9°$N in agreement with previous authors (Kniffka et al., 2019). The domain is bounded at its southern part by the gulf of Guinea and at its northern part by the sudanian savanna and desertic areas of Sahel and encompasses guinea savanna and forest ecosystems. Major conurbations are located on the shore of the gulf of Guinea: Abidjan (Côte d'Ivoire), Accra (Ghana), Lomé (Togo), Cotonou (Bénin) and Lagos (Nigeria). We have collected observations at 3 coastal locations, namely Abidjan, Cotonou and Koforidua, and 4 inland locations, namely Savè, Lamto, Ilorin and Comoé, (see Table 1 for geographical coordinates). The sites labelled Abidjan and Cotonou are respectively located downtown the city of Abidjan ($\approx$ 4.4 million inh.) and the conurbation of Cotonou ($\approx$ 1,7 million inh. including satellite cities). Savè site is located in the medium-sized city of Savè ($\approx$ 90,000 inh.). Lamto is a rural remote site located 200 km north of Abidjan. Comoé site is located near the village of Nassian at the southern edge of La

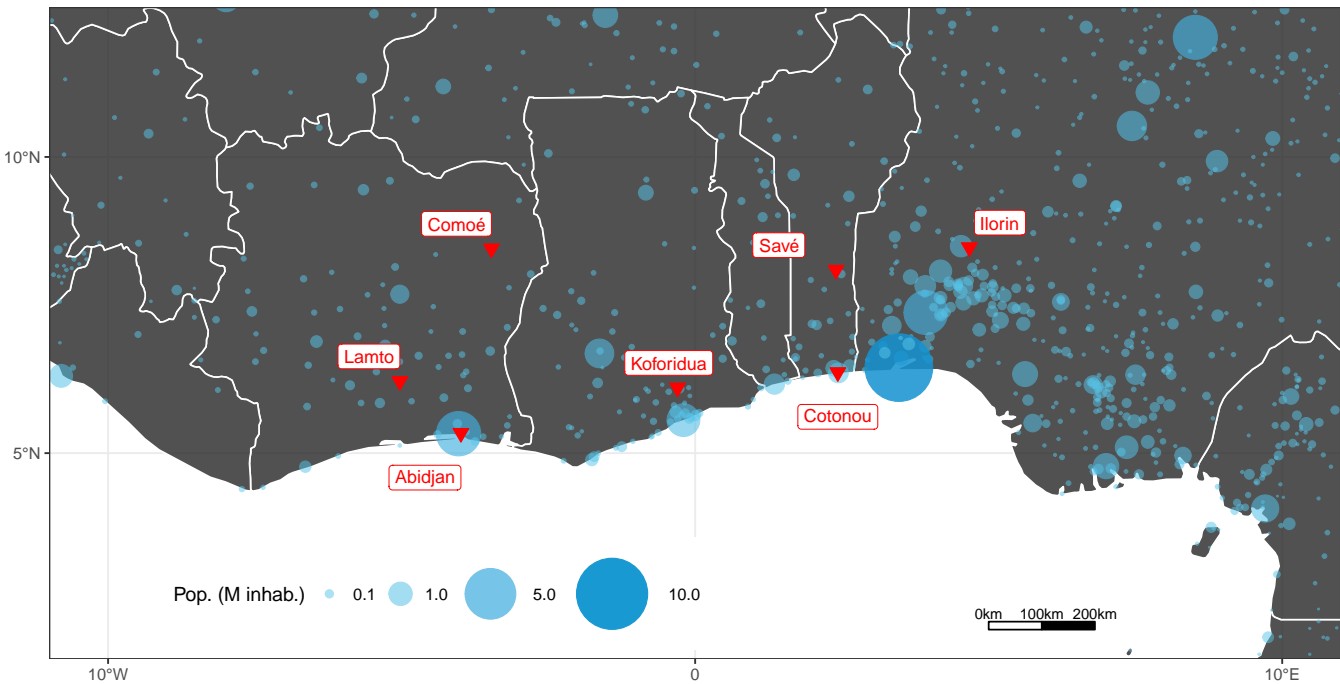

**Figure 1.** Map of south Western Africa (SWA) indicating the (red triangles) geographical locations of the sun photometer sites and (blue circles with legend) the location and population of urban areas having more than 1,000 inhabitants (source https://public.opendatasoft.com/).

Comoé natural reserve. Ilorin site is located at the Department of Physics on the campus of the University of Ilorin ($\approx$ 800,000 inh.) in Nigeria. Koforidua site is located at the main campus of All Nations University College about 5 km from Koforidua City ($\approx$ 120,000 inh.), 50 km north of Accra, Ghana.

## 2.1 Sun photometers

Table 1 summarizes the location, type of instrument and observation periods. We have used different types of sun photometers, automatic and handhelds. The automatic CIMEL sun photometer is the reference instrument used in the AERONET network (Holben et al., 1998) for measuring the AOD and retrieve columnar aerosol optical properties and size distribution. We have used the level 2 quality assured daily averages processed with the version 3 of the aerosol optical depth algorithm (Giles et al., 2019). We used the data for Ghana (station named Koforidua_ANUC located at 6° 6' N, 0° 6' W), Nigeria (station named Ilorin located at 8° 29' N, 4° 40' E) and Côte d'Ivoire (station named Lamto located at 6° 13' N, 5° 2' W). The geophysical station of Lamto was early equipped in 1997-1998 then the automatic sun photometer was restored back in 2017.

Handheld sun photometer is a well-known scientific instrumentation for measuring atmospheric transmission (Porter et al., 2001; Volz, 1959, 1974). The first type of handheld photometer we used is the one manufactured by CIMEL, hereinafter called HHC. HHC was operated during 2 years between April 2006 and March 2008 at Lamto geophysical station. The operating wavelengths are 440, 670 and 870 nm. The second handheld sun photometer is a new lightweight instrument manufactured by

TENUM (http://www.calitoo.fr) and named CALITOO (Djossou et al., 2018). CALITOO operating wavelengths are 465 nm, 540 nm and 619 nm. The sun photometer measures the Sun irradiance at the 3 wavelengths so no additional check on the AOD curvature (Kaskaoutis and Kambezidis, 2008; Sharma et al., 2014) can be applied however the spectral consistency between the AODs (observed at 540 nm and computed using the Ångström exponent) is checked. The atmospheric optical depth is then retrieved following the Beer-Lambert law knowing the calibration constant for each instrument and the relative air mass. The AOD is then retrieved after subtracting the Rayleigh and trace gases optical depth.

For the HHC, observations were acquired twice a day at around 9:00 and 15:00 UTC. For the CALITOO sun photometer, the observations were acquired at around 13:00 LT. The operators were asked to make measurements only when the sun was not obscured by clouds and have proceed with a sequence of 5 measurements within about 15 minutes. The presence of sub-visible cirrus or broken clouds within the field of view induces spurious variation in the atmospheric transmission (Smirnov et al., 2000) that can be easily detected by looking at the standard deviation of the 15-minute series of AOD measurements. An arbitrary threshold of 0.2 on the standard deviation has been selected to remove the cloud-contaminated observations. The diurnal variability range is expected to be less than 10% for our site conditions (Smirnov, 2002). The sun photometer observations are reported as daily averages.

The total uncertainty in AOD for the AERONET instruments is $\pm 0.01$ for $\lambda > 440 nm$ and $\pm 0.02$ for shorter wavelengths (Holben et al., 1998). CALITOO sun photometers were calibrated prior to the site deployment using the Langley-plot method (Soufflet et al., 1992; Schmid and Wehrli, 1995) at the Izaña high-altitude observatory (Basart et al., 2009). CALITOO observations were compared to coincident AERONET observations before and after the field experiment. The total uncertainty in AOD is estimated to $\pm 0.02$ for all the wavelengths. Post-field measurements indicates a change of about 1% per year in the calibration.

AOD measurements are all reported at 550 nm because this wavelength is a reference for visibility calculation (Boers et al., 2015) and satellite mission (e.g. Remer et al., 2008). The Ångström exponent (AE) (Ångström, 1961) is computed between wavelengths 465 and 619 nm for the CALITOO and 440 and 670 nm for the HHC, and between 440 and 675 nm for the AERONET.

## 2.2 Satellite data

The Moderate Resolution Imaging Spectroradiometer (MODIS) aerosol products (Remer et al., 2005, 2008) have been widely used by the scientific community for assessing the impact of aerosols on global climate (Boucher et al., 2013) or air quality (van Donkelaar et al., 2016). The MODIS AOD is also used in operational data assimilation for weather forecast (Benedetti et al., 2009; Lynch et al., 2016). The MODIS Level 2 product has a spatial resolution of 10 km × 10 km at Nadir but increasing to 20 km × 40 km at the edges of the swath. The MODIS Level 3 is a regular gridded product having a spatial resolution of $1° × 1°$. Most of the time, the validation exercise of MODIS derived aerosol parameters consists in a comparison between sun photometers observations and MODIS L2 pixels co-located in space and time (Remer et al., 2020). A box of 5 × 5 MODIS pixels and a time slot of ± 30 min the satellite overpass is admitted to be a good compromise, however the window-size dependence is small and this compromise is more dictated by statistics rather than physics (Ichoku et al., 2002).

Gridded daily or monthly mean MODIS level 3 AODs have also been demonstrated to fit the AERONET retrievals (Ruiz-Arias et al., 2013; Wei et al., 2019). As the objective of the paper is to address the ability of MODIS data to reflect aerosol changes in a specific area rather than the validation of retrieval algorithm, hereinafter we rely on the gridded level 3 MODIS product of AQUA satellite (namely MYD08_D3) from 2003 to 2019. MODIS AQUA has been selected following the recommendations of Wei et al. (2019) for long-term trend analysis. This product provides several values for the AOD depending on the underlying surface and the algorithm used. For a sake of consistency between the different sites, we use the product named AOD_550_Dark_Target_Deep_Blue_Combined_Mean from the version 6.1 (Levy et al., 2013) of the MODIS processing algorithm, which is a combination of the "Dark target" (Levy et al., 2010) and "Deep blue" (Sayer et al., 2013) methods. For the coastal sites, both AOD over land (namely Aerosol_Optical_Depth_Land_Mean or Deep_Blue_Aerosol_Optical_Depth_550_Land_Mean) and over ocean (Aerosol_Optical_Depth_Average_Ocean_Mean) are also provided. We use the "Deep Blue" AE (namely Deep_Blue_Angstrom_Exponent_Land_Mean) and compute an Ångström exponent from the spectral AODs over land and ocean, respectively. For the purpose of satellite validation, the satellite AOD and AE of the nearest cell to the photometer location are extracted. We have adopted the evaluation metrics proposed by Sayer et al. (2014) including the linear correlation coefficient, the median bias, the root mean square error, the mean absolute percentage error, and the fraction of data falling within the MODIS expected error (EE) given by $EE = \pm(0.05 + 0.15 \times AOD)$.

## 2.3 Surface concentration observations

From February 2015 to March 2017, Abidjan and Cotonou were equipped with PM2.5 monitoring stations (Djossou et al., 2018). Particles were collected on 47 mm diameter filters (quartz and PTFE filter types) at a flow rate of 5 L/min. Samplers were equipped with a PM2.5 mini Partisol impactor. PTFE filter were weighted before and after the sampling with a microbalance Sartorius MC21S.

The total volume of filtered air is measured by a GALLUS-type G4 gas meter. Mass concentrations of PM2.5 are estimated from the mass load of particles on the filters and the total volume of air. The exposure duration of the filters is one week. A period of one week is sufficient to capture the main temporal pattern of atmospheric aerosols over a long period of time (Ouafo-Leumbe et al., 2017) and has been selected as a trade-off between logistics and observations.

## 3 Sun photometer results

### 3.1 Daily statistics

A total of 2323 handheld sun photometer observations (including data collected during the 2006 campaign) have been acquired. Starting and ending dates are reported in table 1 along with the number of observations, median and interquartile range (IQR) of AOD and AE distributions. We select the AERONET data until the end of the CALITOO observation period, i.e March 2017 for a total number of 1248 daily observations. There is an excellent time coverage for the stations of Lamto and Cotonou by the CALITOO observations. Observations were performed for 66% of the time in Cotonou, and 68% in Lamto. As a comparison,

this rate is 68% for the automatic sun photometer in Koforidua, indicating that handheld measurements can be as representative as the automatic ones. This rate drops to 24% in Abidjan, 39% in Savè due to operating issues leading to gaps in the time series.

Considering all the stations, the AOD ranges between a minimum of 0.07 and a maximum of 3.76. The highest AOD acquired by the CALITOO instrument is 3.50 in Cotonou in March 2015 and the highest AOD recorded by the AERONET sun photometer is 3.76 in Ilorin in Dec. 2016. The median AOD ranges between a minimum of 0.47 at Lamto and maximum of 0.66 at Comoé. Considering all the daily measurements for all the sites, the median AOD is 0.52, IQR=(0.33, 0.82). AOD observations at Cotonou and Abidjan are rather similar, with a median AOD=0.55 (0.38, 075) and 0.58 (0.35, 0.86), respectively.

The observations with the automatic sun photometer in Koforidua shows AOD in the same range as the 2 aforementioned stations, with a median AOD=0.49 (0.33, 0.84). The observations performed at Lamto with the 3 kinds of sun photometers show similar AOD: 0.47 (0.3, 0.72), 0.55 (0.35, 0.80) and 0.56 (0.38, 0.85) for CALITOO, HHC and AERONET respectively. The difference in the statistics for the 3 instruments at Lamto is due to different sampling periods (see Table 1) although the coincident observations between CALITOO and AERONET in 2017 shows an excellent agreement (see below).

The median AE is between a minimum of 0.33 (0.13, 0.55) at Comé and a maximum of 0.78 (0.56, 1.09) at Koforidua. Observations performed with CALITOO at Lamto shows a slightly lower range of AE values than the ones performed with the HHC. The difference between CALITOO median AE and HCC median AE is -0.09. However it should be noted that the statistical distribution of AOD values has an impact on the corresponding AE distribution (Wagner et al., 2008). AOD observed at Lamto during the HHC period (2006-2008) being higher than the ones observed during the CALITOO period (2014-2016),

it is not expected to have the same AE values. Nonetheless, the difference is within the expected accuary.

    Coincident AERONET and CALITOO observations (N=31) were acquired between January and March 2017 at Lamto (see Figure 3 and Table 1). Figure 2 shows the scatter plot for the corresponding daily AOD and AE There is an excellent agreement for both AOD (regression coefficient R=0.93) and AE (R=0.87) between the two instruments.

**Table 1.** Summary of observations period, number of days of observations (N) per instrument and location. Median and interquartile range for aerosol optical depth (AOD) and Ångström exponent (AE).

| Site | Type | Latitude | Longitude | period | N | AOD median (IQR) | AE median (IQR) |
|------|------|----------|-----------|--------|---|------------------|-----------------|
| Lamto | HHC | 6°13' N | 5° 2' W | Mar. 2006- Mar. 2008 | 524 | 0.55 (0.35, 0.80) | 0.68 (0.42, 0.96) |
| Abidjan | CALITOO | 5° 20' N | 3°59' W | Feb. 2015- Apr. 2017 | 190 | 0.55 (0.38, 0.75) | 0.73 (0.44, 0.97) |
| Lamto | CALITOO | 6°13' N | 5° 2' W | Nov. 2014- Mar. 2017 | 499 | 0.47 (0.30, 0.72) | 0.59 (0.35, 0.86) |
| Save | CALITOO | 8° 01' N | 2° 28' E | Sep. 2015- Oct. 2017 | 411 | 0.61 (0.42, 0.86) | 0.49 (0.26, 0.73) |
| Comoe | CALITOO | 8° 27' N | 3° 28' W | Jan. 2016- Feb. 2017 | 82 | 0.66 (0.43, 0.95) | 0.33 (0.13, 0.55) |
| Cotonou | CALITOO | 6° 22' N | 2° 26' E | Nov. 2014- Jun. 2016 | 615 | 0.58 (0.35, 0.86) | 0.58 (0.32, 0.89) |
| Lamto | AERONET | 6°13' N | 5° 2' W | Jan. 2017- Mar. 2017 | 35 | 0.74 (0.59, 0.83) | 0.82 (0.58, 1.08) |
| Ilorin | AERONET | 8° 29' N | 4° 40' E | Jan. 2014- Mar. 2017 | 472 | 0.52 (0.30, 0.89) | 0.63 (0.39, 1.00) |
| Koforidua | AERONET | 6° 6' N | 0° 6' W | Dec. 2015- Mar. 2017 | 264 | 0.54 (0.32, 0.92) | 0.78 (0.56, 1.09) |

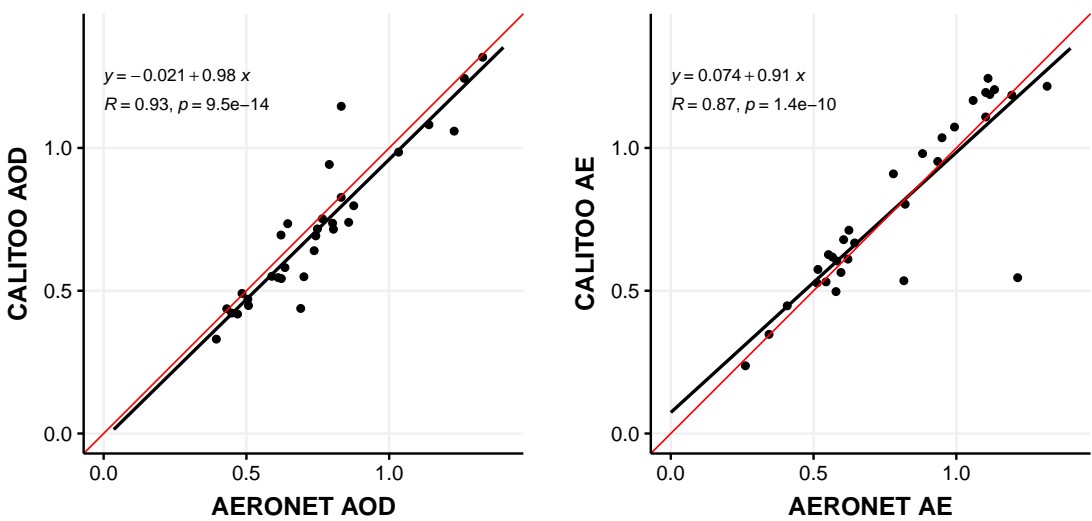

**Figure 2.** Scatter plot of AOD and AE observed at Lamto by CALITOO and AERONET instrument between January and March 2017 (number of points = 31).

## 3.2 Time series

The daily AODs and AEs for each site and each instrument between 2015 and 2017 (between 2006 and 2008 for HHC) are presented in Figure 3 and 4, respectively. A similar seasonal pattern is observed in the different time series. There is an increase in AOD during the main dry season (December to March) and a decrease during the rainy season (April-July). The 2-week smoothing average reveals a high degree of correlation between time series. The correlation coefficient between Cotonou and Lamto time series is R=0.82, being R=0.90 between Cotonou and Koforidua. During the short overlap period in March 2017,

the CALITOO and AERONET instruments shows similar AOD values at Lamto station. The Comoé time series is the weakest one with only 82 data points. The 2006-2008 HHC Lamto time series has the same pattern as the one recorded by CALITOO in 2015-2017 and showing two maxima in the dry season, one in December and another one in January-February.

The seasonal pattern for AEs (Figure 4) shows an opposite cycle with lower values in the dry season and higher values during the rainy season. AE seasonal cycle is clearly affected by the winter dry period with dust laden air masses that decrease

the AE values. The median AE value during the first half of the year (all site except Comoé) is 0.36 (0.23, 0.62) and 0.69 (0.43, 1.00) during the second half of the year. AOD in the last quartile (AOD above 0.82) are mostly (72%) observed during the month of December, January and February and are associated with a median AE of 0.44 (0.24, 0.64). While low AODs (first quartile, i.e. below 0.33) are associated to a median AE of 0.89 (0.61, 1.12) and observed between August and October (51% of the observations). The difference in AOD between the inland and the coastal sites is less than 0.05, with differences up to

0.1 between April and June, the AOD at the coastal stations being higher than inland. AE are higher at the coastal stations than at the inland stations by 0.15 on average, reflecting the influence of urban air pollution at the coastal stations.

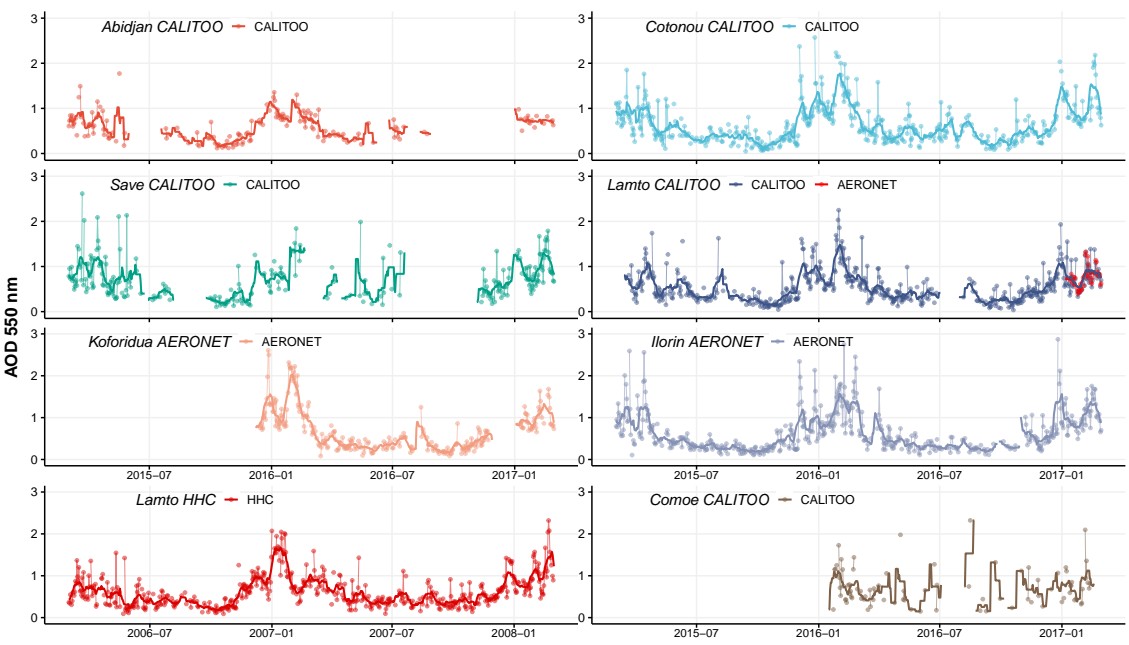

**Figure 3.** Daily Aerosol Optical Depth at 550 nm. The name of the site and the type of instrument used is given in the legend of each plot. Solid line is a 2-week smoothing average.

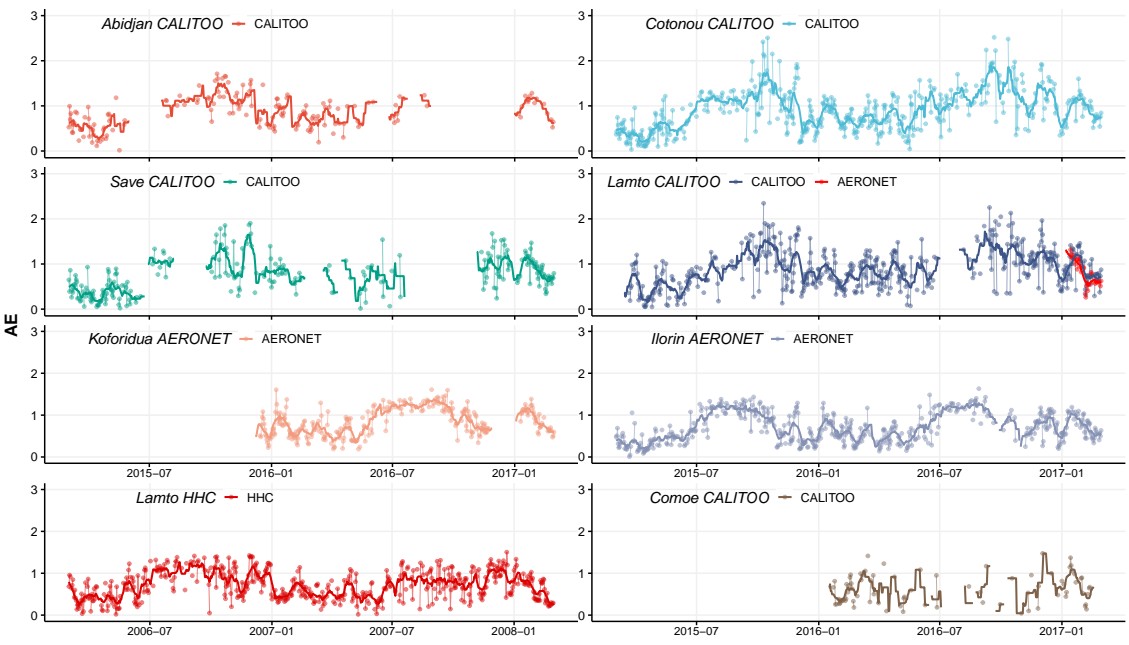

**Figure 4.** Same as Figure 3 for the visible Ångström exponent.

### 3.3 Comparison with MODIS aerosol products

Table **??** gives the statistics of the regressions for each site and per instrument presented in Figure 5. We have then adopted a log-log representation on the scatter plots presented in Figure 5 as the AOD distribution has a significant right skewness (O'Neill et al., 2000). Figure 5 presents also the MODIS expected error (EE, blue lines). Whatever the site is, there is a significant correlation between the MODIS and sun photometer observations. The Pearson correlation coefficient R ranges between 0.75 (Comoé) and 0.94 (Koforidua). For the CALITOO observations, R is between 0.75 and 0.90 (Cotonou). The lowest RMSE values are found for the measurements operated using the CALITOO at the coastal sites of Abidjan and Cotonou. The MAPE is on average 30%. The sites in Cotonou and Abdijan are not biased and present a fraction of data falling within the MODIS EE above 60%. All the inland sites are biased and it results in a rather low fraction of data falling within the MODIS expected error.

**Table 2.** Results of the MODIS and sun photometer AOD comparison by site location and type of instrument indicating the (N) number of data, (R) Pearson correlation coefficient, (RMSE) root mean square error, bias, (MAPE) mean absolute percentage error and (fEE) fraction falling within the MODIS expected error.

| site | instru | N | R | RMSE | Bias | MAPE | fEE |
|------|--------|-----|------|------|-------|------|-----|
| Cotonou | CALITOO | 401 | 0.88 | 0.22 | -0.02 | 24 | 64 |
| Save | CALITOO | 254 | 0.79 | 0.31 | -0.19 | 35 | 35 |
| Abidjan | CALITOO | 118 | 0.86 | 0.14 | -0.02 | 18 | 76 |
| Lamto | CALITOO | 185 | 0.86 | 0.26 | -0.15 | 29 | 50 |
| Comoe | CALITOO | 47 | 0.76 | 0.37 | -0.22 | 32 | 44 |
| Ilorin | AERONET | 264 | 0.91 | 0.32 | -0.19 | 33 | 39 |
| Koforidua | AERONET | 144 | 0.93 | 0.30 | -0.20 | 26 | 46 |
| Lamto | AERONET | 17 | 0.87 | 0.34 | -0.32 | 50 | 0 |
| Lamto | HHC | 181 | 0.83 | 0.39 | -0.29 | 37 | 25 |

The bias has a seasonal behavior being highest during the dry season between December and March. An underestimation of the MODIS AOD is then observed at maximum in January with an absolute bias of -0.33 (39% in relative) at the inland sites. Sayer et al. (2014) have already pointed out the possible differences in the "Dark Target" and "Deep blue" algorithms. It appears from the Figure 6 in Sayer et al. (2014) that the dry to humid savanna transition zone in sWA is an area where large differences exist in both retrieval techniques during the dry season. Those differences can explain that the "Merge" product used in this study has a large bias during the dry season in the northern part for the inland sites. So the North-South AOD gradient in this area remains difficult to assess based on satellite products.

For all the sites considered in this study and whatever the sun photometer is, the correlation between MODIS AE and sun photometer AE is non significant. This finding is coherent with the results of Antuña-Marrero et al. (2018) and Sayer et al. (2013). The histogram of the Sun photometer and MODIS-derived AE are presented in Figure 6. The default values AE=1.5 and AE=1.8 in the MODIS AE Deep Blue product have been removed (Sayer et al., 2013). The MODIS median AE is biased

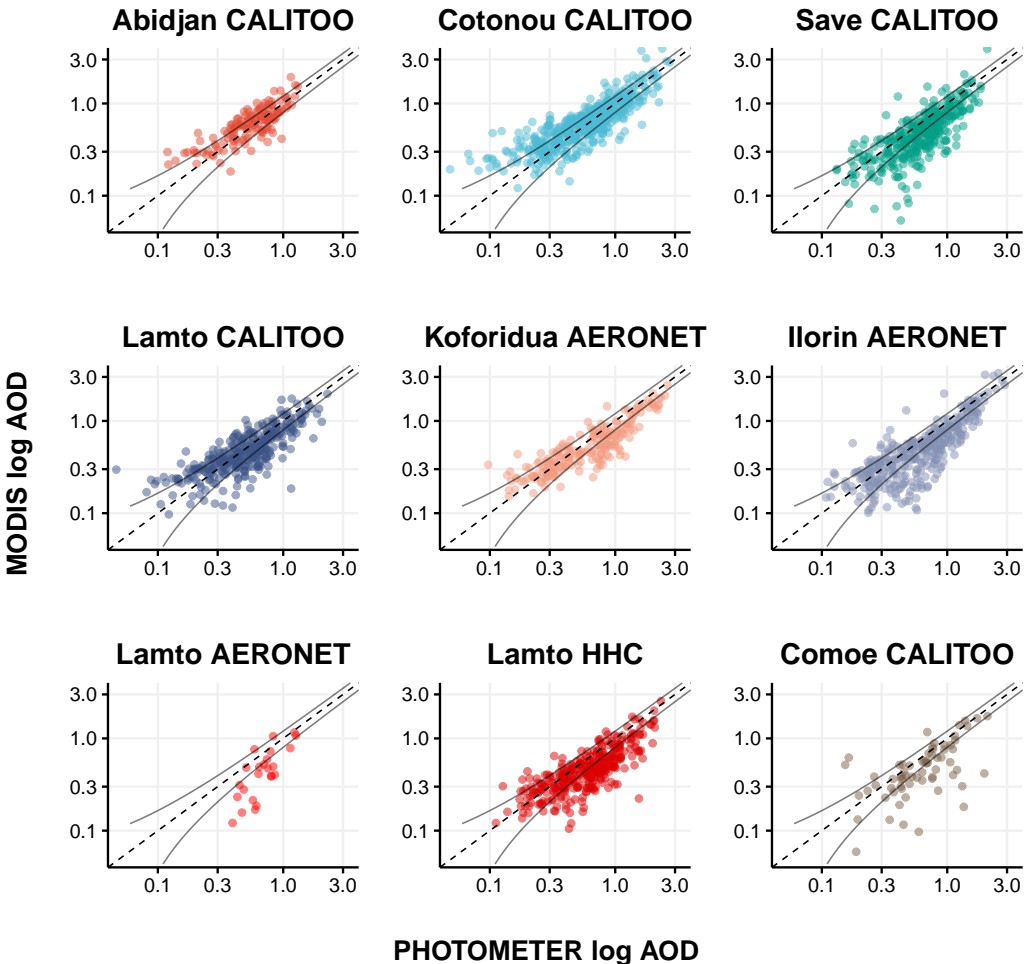

**Figure 5.** Scatterplots of MODIS versus sun photometer AOD for the 3 types of sun photometers (automatic AERONET, and handheld CALITOO and CIMEL) and different sites (Lamto, Comoé, Savè, Cotonou, Abidjan, Ilorin, Koforidua)

.

by 0.32 and the distribution of AE values doesn't follow a normal distribution, whereas the distribution of AE values for the sun photometers does. At the coastal sites, the MODIS AE Ocean algorithm reproduces well the left side of the histogram (lowest AE) while it significantly underestimates AE above 0.8. MODIS AE Land algorithm at 0.5 indicates that a dust-like aerosol model has been selected (Levy et al., 2010) however there is no systematic association with low sun photometer AE. However it could be possible to adapt the interval bounds (lower and upper limits of AE values) for each category, the statistical distribution of MODIS AE values doesn't fit the sun photometer ones and could lead to misclassification of daily observations.

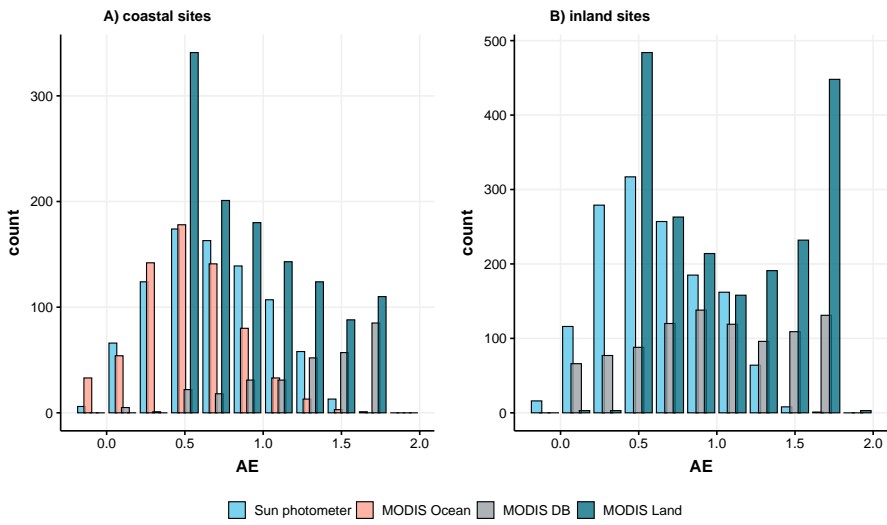

**Figure 6.** Comparative histograms of Ångström exponents for coincident sun photometer, MODIS Deep Blue, MODIS Land and Ocean algorithms, for (A) coastal sites and (B) inland sites.

# 4  Aerosol type and relationship with surface concentrations

## 4.1  Aerosol type

AE is an intensive aerosol optical parameter and depends on the spectral aerosol extinction coefficient (Nakajima et al., 1996; Eck et al., 1999; Holben et al., 2001). AE is influenced by the aerosol size distribution and is commonly used to identify aerosol types (Léon et al., 1999; Kaskaoutis et al., 2009; Perrone et al., 2005). Aerosol types having a dominant fraction of their size distribution in the coarse mode, like dust and sea-salt particles, are associated with a lower value of AE than aerosol types having a size distribution dominated by the accumulation mode, like secondary and combustion aerosols. The concurrent changes in AOD and AE help to distinguish generic aerosol types in sun photometer time series (Toledano et al., 2007; Verma, 2015). Mineral dust tends to increase atmospheric AOD and decrease AE (Hamonou et al., 1999) while biomass burning events tends to increase both AE and AOD (Eck et al., 2003).

The AOD versus AE scatter plot can be used to cluster the observations by aerosol broad categories corresponding to a main source, like coarse mineral dust of biomass burning aerosols. The thresholding in AOD and AE for aerosol type identification varies from one site to another and also depends on the distance from aerosol sources upwind the site (Verma, 2015; Benkhalifa et al., 2017). In particular, the classification based on AOD vs. AE values is incapable of determining aerosol absorption properties (Giles et al., 2012; Cazorla et al., 2013). Figure 7 presents the scatter plots of AODs (log scale) versus AEs for each site and splitted by seasons. We have considered 4 seasons corresponding to the long dry season (Dec.-Mar), the long wet season (Apr.-Jun.) and the short dry ( Aug.-Sep.) and short wet season (Oct.-Nov.). For the sites with the most comprehensive data set over the different seasons (Lamto, Cotonou, Koforidua, Ilorin) the AOD vs. AE plots show a similar pattern with

decreasing AODs almost linearly as AEs increases. The lowest AEs are observed during the long dry season and are associated to largest AODs indicating the presence of coarse mineral dust. The presence of dust can be also observed in the wet long season. During the short dry season, all the sites excepted Comoé show larger AEs and lower AODs than for the other seasons.

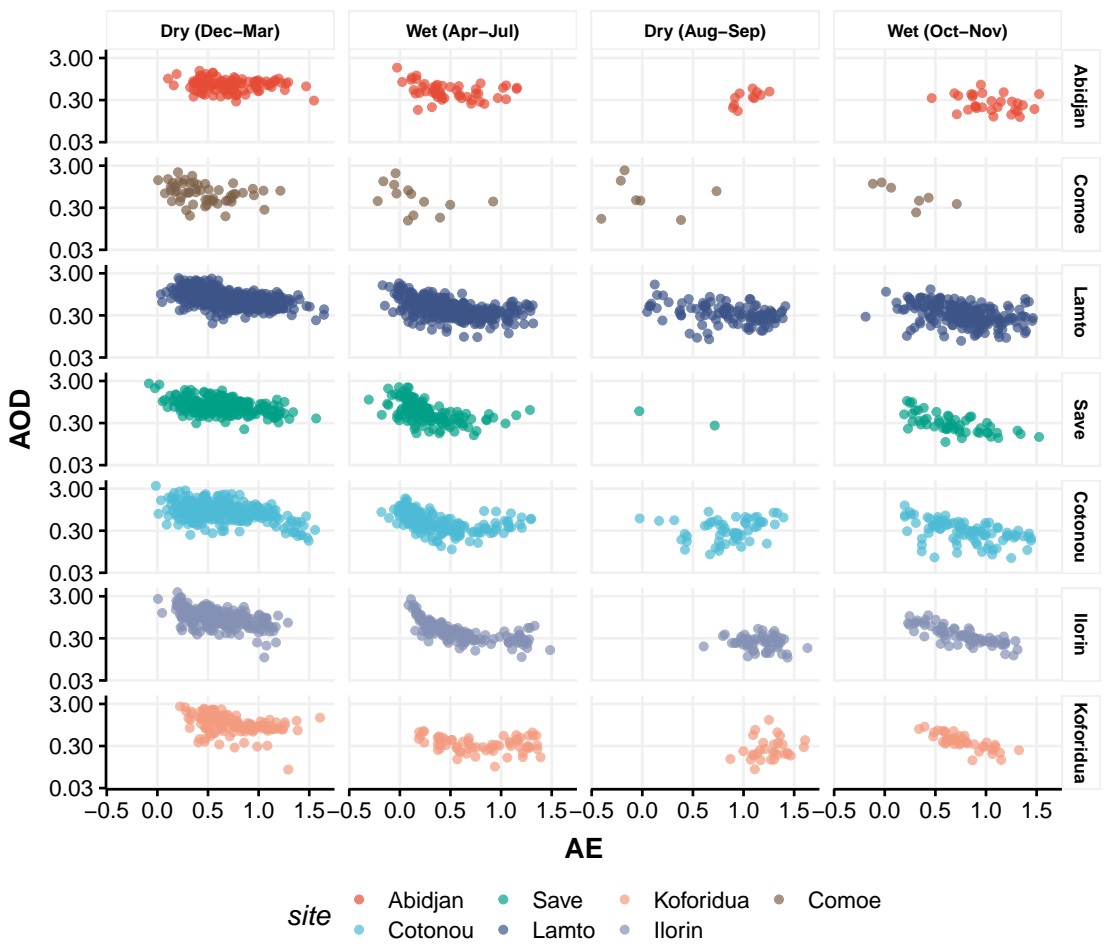

**Figure 7.** Scatter plots of sun photometer aerosol optical depth (AOD) versus Ångström exponent (AE) splitted by sites and by seasons.

It can be noticed from Figure 7 that there is no clear definition of AOD and AE thresholds for each aerosol categories and the scatter plots of Figure 7 reflects the high mixing of different aerosol types. The absolute error on AE is a function of the relative error on AODs and depends on the spectral range investigated (Hamonou et al., 1999; Wagner et al., 2008). Typical error on AE is ± 0.3 for an AOD of 0.2 and there is a risk of over interpretating AE variations.

       In this paper we classify the daily observations according to the AE values using a simple statistical analysis and a threshold
on AOD. The whole sun photometer dataset is divided into 3 quantiles. The first third corresponds to AE≤0.45 and observa-

tions having an AOD $\geq$ 0.8 are labelled "coarse dust", while observations having an AOD<0.8 are labelled as "mixed". The threshold on AOD corresponds to the third quantile of AOD distribution and is used to better identify dust events. The last third corresponds to AE$\geq$0.80 and is labelled "urban-like". The data having 0.45<AE<0.80 falls into a "mixed" category, being more populated than the two others. This rather crude classification enables to identify the main aerosol influence with a significant number of observations in each category.

**Table 3.** Percentage of daily observations in the aerosol categories at each site.

| site | Coarse dust | Mixed | Urban-like |
| --- | --- | --- | --- |
| Abidjan | 3.7 | 56.8 | 39.5 |
| Comoe | 22.0 | 69.5 | 8.5 |
| Lamto | 8.6 | 56.5 | 34.9 |
| Save | 12.2 | 67.9 | 20.0 |
| Cotonou | 10.2 | 57.6 | 32.2 |
| Ilorin | 12.7 | 50.6 | 36.7 |
| Koforidua | 4.9 | 47.3 | 47.7 |

Table 3 presents the typology of the sites according to the aforementioned classification. Comoe is the most influenced by coarse dust aerosol (20%), followed by Ilorin (12.7%), Savè (12.2%), Cotonou (10.2%) and Lamto (8.6%). The southwestern sites, Abidjan (3.7%) and Koforidua (4.9%) are less influed by dust events than the eastern sites. As expected from Figure 7 all the site shows 'urban-like' category that also corresponds to low AODs. Two sites are less influenced by urban-like aerosols than the others, namely Save (20%) and Comoe (8.5%). For the other sites, the 'urban-like' category ranges between 47.7% (Koforidua) to 32.2% (Cotonou).

As SWA lacks dedicated studies on aerosol characterization, there are few other data to compare with. Hamill et al. (2016) have proposed a sophisticated aerosol classification for Africa based on AERONET observations however none of the sites are located in our area. Hamill et al. (2016) have classified Djougou (Northern Benin located at 9° 42' N) as a dust site that is seldom affected by biomass burning. Savè and Ilorin are located around 200 km south of Djougou and the influence of dust is still significant compared to the coastal sites. Comoé is located 600 km westward of Djougou and probably less influenced by the dust transport from the Bodélé area in Chad however measurements acquired at Comoé don't cover a full season and the exact frequency of dust or biomass burning events remains uncertain.

## 4.2 Relationship to surface concentrations

The changeover between the monsoon and the harmattan results in a change in the vertical distribution of aerosol layers and in the type of aerosols Djossou et al. (2018). The harmattan flow carries continental aerosols in the lowest part of the atmosphere during the dry winter season (December to March). During the dry winter season the days with high AOD are often associated with an increase in the PM2.5 surface concentration leading to a high correlation coefficient between AOD and PM2.5.

The correlation coefficient between weekly mean AOD and PM2.5 measured in Cotonou and Abidjan is R=0.75 (N=105) when considering the whole observation period. The correlation coefficient can reach R=0.96 (N=6) during specific aerosol events observed from December 2015 to January 2016 in the heart of the dry season. During other periods of the year, the correlation remains weak because the concentrations are less fluctuating than during the winter period.

PM2.5/AOD ratios are estimated using the daily AOD observations and the weekly PM2.5. The PM2.5/AOD is basically the amount of PM2.5 that is expected per unit of AOD. It was first promoted by van Donkelaar et al. (2010) as a conversion factor (Zheng et al., 2017; Yang et al., 2019). The PM2.5/AOD ratio reflects how a change in the AOD affects the ground surface concentrations, however there is no evidence of a unique relationship between both quantities. The PM2.5/AOD ratio depends on the vertical stratification of the aerosol layers in the atmosphere due to mixing processes in the boundary layer or large scale advection (Sayer et al., 2016). The ratio depends also on the aerosol size distribution and chemical properties that are changing during the transport and the aging of the aerosols.

In the specific case of the coastal cities of the Gulf of Guinea, we are interested in evaluating how the change in aerosol type during the season, and in particular the seasonal advection of mineral dust from the desert area, may affect the PM2.5 surface concentrations. For this purpose, we estimate a PM2.5/AOD ratio per aerosol type and per season.

Each daily AOD observation is associated to an aerosol type (Coarse dust, mixed or urban-like) depending on the corresponding daily AE value. The daily AODs are associated to the corresponding PM2.5 observation using equation 1.

$$PM2.5_{weekly} = \frac{1}{n}\sum_{i=1}^{n}(\sum_{t=1}^{3}\beta_{t,s}\tau_{i,t,s}) \tag{1}$$

The corresponding PM2.5/AOD coefficients, $\beta_{t,s}$ in equation 1 where $t$ represents the aerosol type and $s$ the season, are evaluated using a multilinear regression on the observations collected in Cotonou and Abidjan for each season independently. Cotonou and Abidjan samples are pooled together to increase the statistical significance and to retrieve average coefficients at the regional level. As the season are not equal in length and the number of observations differs in Abidjan and Cotonou, the number of samples differs, ranging between 71 samples during to long dry season to 24 samples during the dry short season. The significance of the regression and standard error on the coefficients depends on the number of samples. None of the weeks in the short dry period are affected by dusty days, so the coefficient for coarse dust is not retrieved for this period. During the short wet season, only 2 weeks over 26 have a dust contribution and the coefficient is not significant. The PM2.5/AOD ratios by aerosol category are presented in Figure 8 as a function of the season and with their respective uncertainties. The average PM2.5/AOD ratio without accounting for the aerosol category for a given season is also reported. The uncertainties corresponds to the standard error of the coefficients found by regression. The standard error depends on the occurrence of an aerosol category and its relative weight in a given season. For all the season the coefficients for each aerosol type are significant (p<0.05) except for the coarse dust category during the short dry (no data) and the short wet season. The resulting adjusted coefficient of determination for the regression is between 0.76 (long dry season) and 0.83 (wet short season).

The PM2.5/AOD ratio for Coarse dust aerosols ranges between $54 \pm 8$ $\mu g/m^3/AOD$ in the dry long season to $20 \pm 4$ $\mu g/m^3/AOD$ in the long wet season. The seasonal changes in the PM2.5/AOD ratio for coarse dust reflects well the vertical

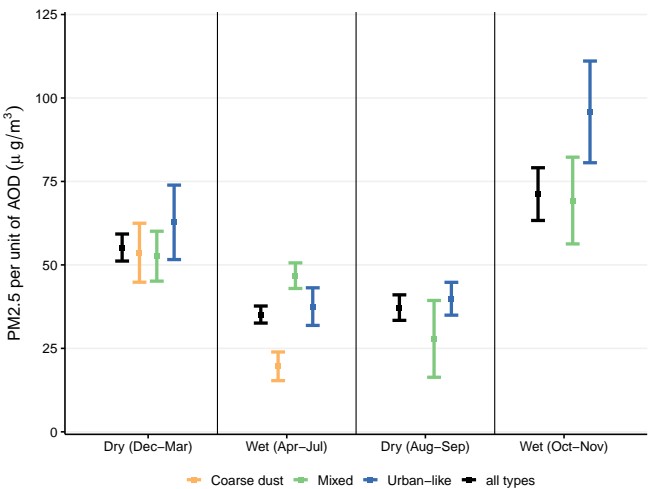

**Figure 8.** Ratio of PM2.5 to AOD for each of the aerosol types during the long dry (Dec.-Mar) and long wet (Apr.-Jun.) seasons and the short dry (Aug.-Sep.) and short wet (Oct.-Nov.) seasons. Data are collected in Abidjan and Cotonou cities from 2015 to 2017.

shift of the dust layer between the dry and the wet season. During the wet (Apr-Jul) season, the air masses are uplifted by the monsoon flow. PM2.5 concentrations remain moderate (21 $\mu g/m^3$ in April) while AODs are still significant (0.57 on average in April) due to the aloft transport. The impact of coarse dust on PM2.5 is higher during the dry season (higher ratio and high 320 AODs) when the dusty air masses are advected close to the ground surface.

The PM2.5/AOD ratio for Mixed aerosols ranges between 53 ± 7 $\mu g/m^3/AOD$ in the dry long season to 27 ± 11 $\mu g/m^3/AOD$ in the short dry season. During the short dry season, only 30% of the weeks are are affected by mixed type aerosols, the remaining being classified as urban-like. The corresponding PM2.5/AOD ratio for mixed type aerosol is close to the one found for dust during the previous season (long wet) indicating the aloft dust transport can be still active but incorrectly 325 classified to the mixed aerosol type due to a low intensity (small AOD).

The PM2.5/AOD ratio for Urban-like aerosols ranges between 96 ± 15 $\mu g/m^3/AOD$ in short wet season to 37 ± 5 $\mu g/m^3/AOD$ in the long wet season. The PM2.5/AOD ratio retrieved in the long dry season for the urban-like category are affected by a larger uncertainty due to a limited impact of Urban-like aerosol during the long dry season compared to Coarse dust. There is a shift of the PM2.5/AOD ratio toward higher value during the short wet season. The short wet season is a 330 transition period during which the stagnation of air masses over land favors the accumulation of pollutants, and also combustion by-products emitted over Nigeria (Marais et al., 2014).

Figure 9 presents the weekly average AODs, satellite-derived and in situ PM2.5 for both Abidjan and Cotonou. The label attributed to each week corresponds to the aerosol type having the largest mean AOD over the week. The period from March to May is dominated by the coarse dust type and there is a clear shift to urban-like type in June-July. A second period of coarse 335 dust is observed in December (2015 and 2016) and is associated to a significant increase in both AOD and PM2.5. PM2.5

during the dusty period of December rise over $100\,\mu g/m^3$. Another sharp increase is observed in February and is associated to the mixed aerosol type. For both years, the two intense periods (December and February) are separated by an interim period showing moderate PM2.5 and AOD and classified as urban-like aerosols.

On average, satellite-derived PM2.5 are agrees with the in situ PM2.5 observations. Indeed the mean difference between retrieved and observed PM2.5 during the 2015-2016 period is less than 1 $\mu g/m^3$ (3%). The MAE is 14 $\mu g/m^3$ and the RMSE is 21 $\mu g/m^3$. The RMSE found here is within the range of previous studies (Ma et al., 2015; Sinha et al., 2015) for other regions of the world and different algorithms. The very intense periods are underestimated, e.g. the mean difference between retrieved and observed PM2.5 is -51 $\mu g/m^3$ (-70%) in December 2015 in Cotonou and nearly a factor of 2 lower in December 2016. The satellite-derived concentrations in January and in March are overestimated. Despite introducing a characterization of the aerosol type, there is still a clear smoothing effect on the weekly concentrations that results from the adjustment of the regression coefficients on a seasonal basis. Using seasonally adjusted coefficients only rather than the seasonally and aerosol type adjusted coefficients has a limited impact on the comparison and decrease the RMSE by only 2 $\mu g/m^3$ on average. The biggest impact is during the long wet season (20% decrease in RMSE) when a lower PM2.5/AOD coefficient is selected for the identified dust cases.

## 5  Trend in the MODIS-derived PM2.5 time series

We have applied the PM2.5/AOD conversion factors to the daily MODIS AOD observations between 2003 and 2019. As the MODIS AE can't be used to classified the daily aerosol observations, we have applied the mean seasonaly adjusted PM2.5/AOD ratios. The database of daily AOD observations consists in 2675 observations for the area of Abidjan and 3018 for the area of Cotonou, corresponding to about 160 observations per year on average. To increase the number of observations per year, we introduce a new area named SWA located between 7° W and 5° E and between 6° N and 4° N. There is at least one MODIS L3 observation per day in this largest coastal area that encompasses both cities.

Mean retrieved PM2.5 is 28.3 $\pm$ 22.2, 30.5 $\pm$ 24 $\mu g/m^3$ in Abidjan and Cotonou, respectively. Almost all the years have a annual average above the EU target value of 25 $\mu g/m^3$, except 2003, 2013, 2014 and 2019. More than 90% of the daily observations are above 10 $\mu g/m^3$ for both cities. A maximum is observed as high as 300 $\mu g/m^3$ during dust event in winter 2010 in Abidjan. During the long dry season 80% of the days have a value above 35 $\mu g/m^3$ while this number drops to 4% during the short dry season.

The MODIS-derived PM2.5 monthly mean annual cycle given in Figure 10 for both cities and SWA area reflects this large seasonal change in the concentrations. A first period is observed between December to March when concentrations are at highest. During this period, the overall mean PM2.5 value is 47 $\mu g/m^3$, concentrations in Cotonou being higher than in Abidjan (max 11% in January). We observe a large difference in April (18% higher in Cotonou) that is clearly attributed to a change in the contribution of coarse dust(+34% in Cotonou) while the contribution of other types remains the same. This higher contribution of dust during the dry period and even more during the intermediate period over Cotonou could be associated to the higher proximity of Cotonou to major dust sources (Bodélé depression) and preferential advection pathways.

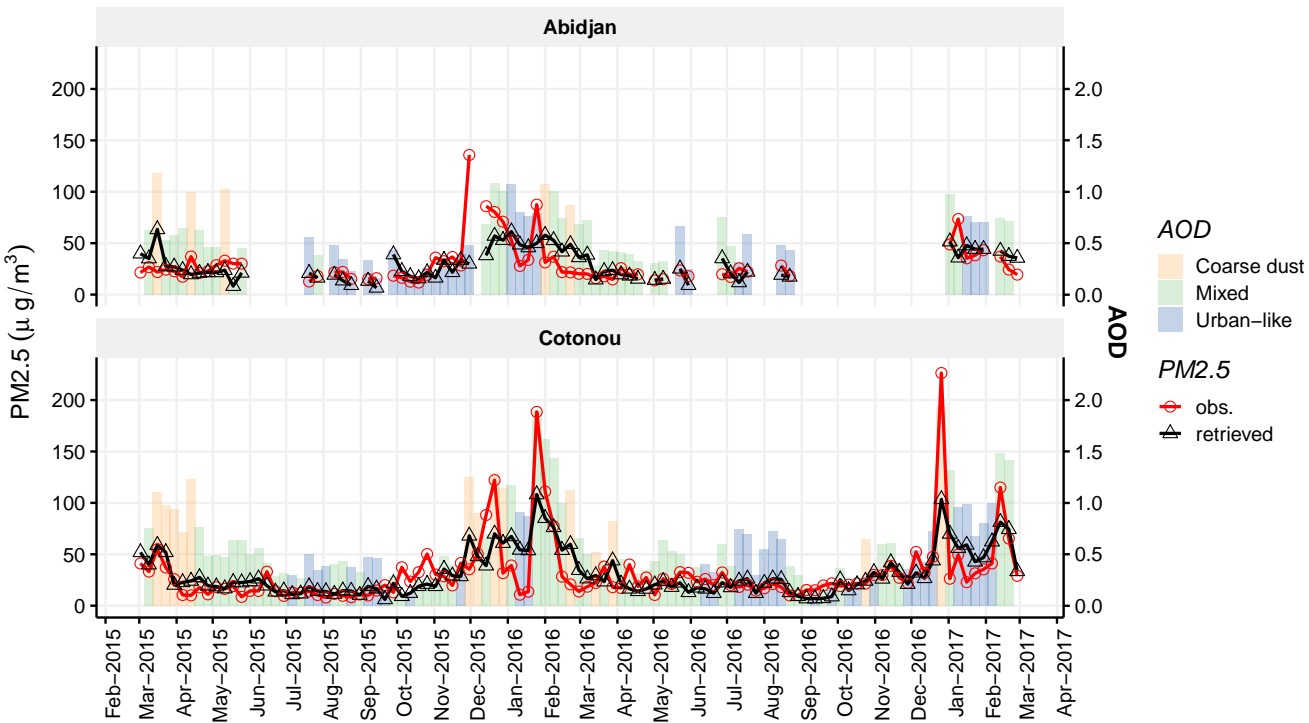

**Figure 9.** In situ and AOD-derived mean weekly PM2.5 from March 2015 to March 2017 in Abidjan and Cotonou. Vertical color bars give the weekly AOD by aerosol category.

A second period is observed between May and September showing mean PM2.5 below 16 $\mu g/m^3$ for both cities and the whole area. The third period corresponds to a steady increase in PM2.5 between September to the December. PM2.5 mean concentration over SWA area is around 11 $\mu g/m^3$ in September and increases up to 37 $\mu g/m^3$ in December, corresponding to an increase by a factor of about 3 in 4 months. A similar increase is observed for Abidjan and Cotonou.

The monthly mean PM2.5 displayed on Figure 11A shows the strong seasonal variation with highest values in January or February every years. The trend on monthly means is retrieved after a seasonal decomposition using a procedure based on Loess (Cleveland et al., 1990). The trend does'nt have an obvious pattern however one can observe a pseudo-cycle of 4 to 5 years. We can notice a decrease in the mean concentrations after 2017. The drop in 2018 and 2019 is due to lower AODs for those two years. The annual mean AOD decreases by 20% between 2017 and 2019 however we didn't investigated further a possible explanation for the decrease. Mann-Kendall'seasonal trend test (Hirsch et al., 1982) applied on monthly means is not significant over the whole 2003-2019 period.

To further investigate a possible trend in the urban-like aerosol, we have selected the data acquired during the short dry period (August-September) during which no dust event have been detected in our sun photometer dataset. Over the period 2003-2017,

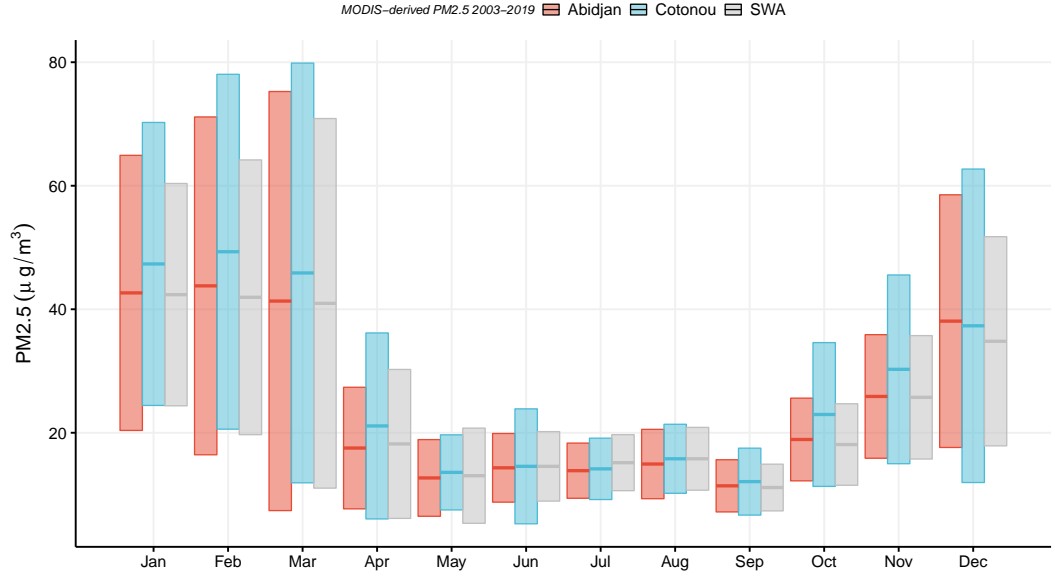

**Figure 10.** Monthly mean annual cycle of MODIS-derived PM2.5 in Abidjan and Cotonou cities and SWA area between 2003 and 2019. Boxes represents the mean ± one standard deviation.

we observe a monotonic trend (Mann-Kendall's tau=0.48) in the annual mean MODIS-derived PM2.5 over the SWA box (Figure 11B). The Thiel-Sen's slope over 2003-2017 is 0.20 with an 95% confidence interval of [0.04, 0.43] corresponding to a monotonic increase in PM2.5 of 3.0 $\mu g/m^3$ over 15 years. The large uncertainty in the observed trend during the short dry period is due to the low PM2.5 concentrations observed during this period (see Figure 10). As PM2.5 are directly linked to AOD, any bias occuring in AOD will affect the PM2.5 concentrations. Moreover the drift in the MODIS AQUA calibration expressed in AOD per decade is 0.01 (Sayer et al., 2019) and will lead to an increase of the same order of magnitude when considering the corresponding PM2.5/AOD conversion coefficient.

## 6   Conclusions

An increase in the anthropogenic emission of atmospheric pollutants is expected as a result of the massive urbanization of the Gulf of Guinea. The scarcity of ground-based observations in sWA is still a limiting factor for a comprehensive understanding of the short-time trend over growing African cities. Moreover, the large influence of natural aerosol emission in sWA produces a complex mixing of particles in the urban atmosphere of sWA cities. In this paper, sun photometer and satellite observations have been used to characterize the magnitude and seasonal behaviour of the aerosol optical depth in sWA. We have set up a small network of lightweight handheld sun photometers, that provides an unprecedented data set on the AOD over sWA between 2015 and 2017. This data set was complemented by additional measurements from AERONET data and observations

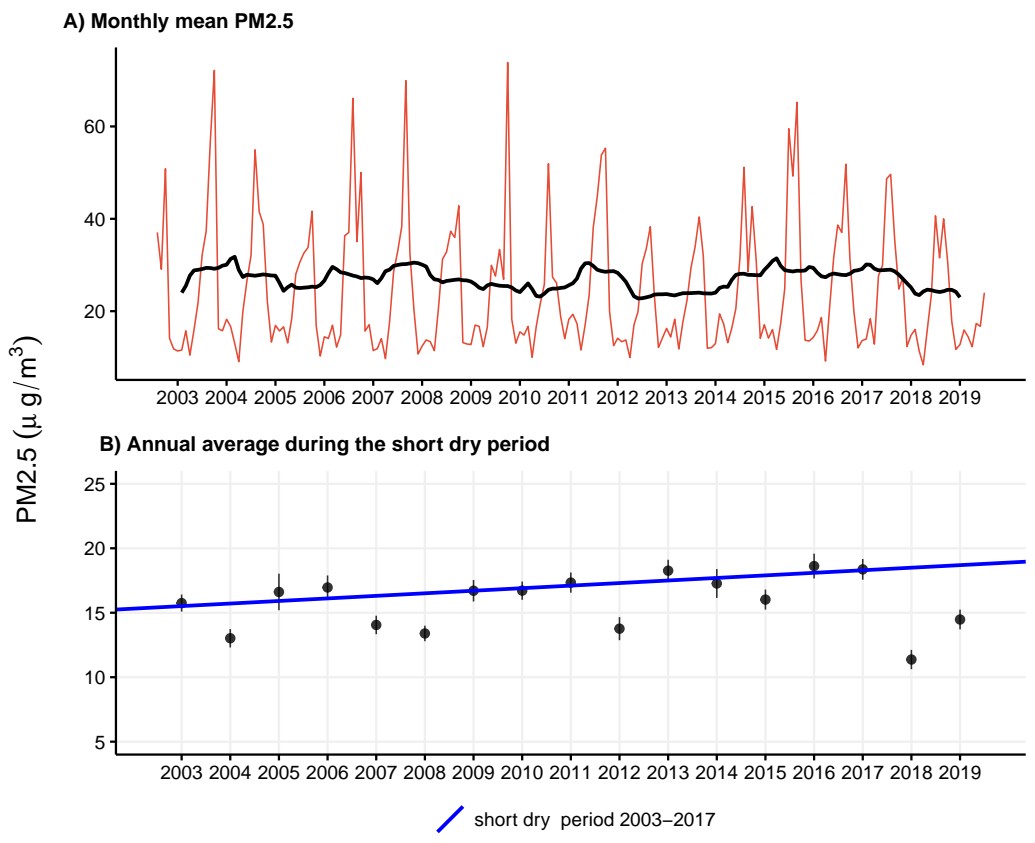

**Figure 11.** MODIS-derived PM2.5 from 2003 to 2019. (A) Monthly mean over SWA and seasonal adjusted trend, (B) annual average during the short dry period and monotic trend computed over 2003-2017.

obtained during a previous campaign in 2006 in Côte d'Ivoire. The comparison of our observations with the MODIS Level 3 gridded satellite observations shows that the satellite AOD derived in the vicinity of the coastal conurbation are excellent, while there is a possible negative bias for the retrievals farther inland, that must be further investigated. Reversely the MODIS

AE doesn't fit the sun photometer observations.

A basic classification using the AOD spectral dependency reveals the large impact of the advection of mineral dust on the AOD seasonal cycle. Dust impacts the cities of the northern part of the gulf of Guinea (namely Abidjan and Cotonou in the present study) from December to May and brings the largest AODs during the months of December and February.

Weekly surface PM2.5 in Abidjan and Cotonou and daily AOD observations were used to estimate a set of AOD to PM2.5

conversion coefficients that accounts for the aerosol category and the season. Despite a good agreement for most of the year, the retrieved PM2.5 underestimates the actual concentrations during the large aerosol events in the dry season. Reversely the PM2.5 are overestimated in early march as a consequence of the shift in altitude of the Harmattan wind. Nonetheless the seasonal variability of the PM2.5 concentrations is in a good agreement with the actual ones.

The seasonal PM2.5/AOD conversion coefficients are applied to the MODIS AOD time series from 2003 to 2019. It was not
possible to adjust the PM2.5/AOD ratio both seasonally and by aerosol type due to the lack of precision in the MODIS AE. No obvious trend is observed in the mean monthly concentrations however trend fluctuates with pseudo period of 4 to 5 years. A link to the 5-year cycle of rainfall in the Sahel (Brandt et al., 2019) could be involved as rainfall is one of main driver of dust emission (Prospero and Nees, 1977) and also as it controls the amount of biomass that can be burnt.

An increase in MODIS-derived PM2.5 is observed over the 2003-2017 period during the short dry period (August-September).
The trend corresponds to an increase of 20% over 15 years. There are several mechanisms that can lead to the increase in the anthropogenic PM2.5 concentrations. Combustion sources are subject to an increase in sWA as well as for the rest of Africa, e.g. organic carbon emissions are multiplied by a factor between 1.5 and 3.0 over 2005-2030 (Liousse et al., 2014). The conurbations of the Gulf of Guinea are under the influence of gas flaring emissions in the Niger delta area (Ologunorisa, 2001). Recent studies show a decrease in gas flaring emissions in the Niger delta area (Deetz and Vogel, 2017; Doumbia et al., 2019)
but the impact of the year-to-year variability of such emissions on regional atmospheric concentrations has to be further investigated. The increase found during the short dry period corresponds to an average annual growth rate of 1.1% being in the lower bound of the emission scenarii however there is no evidence that the observed trend is direclty linked to an increase in the urban emissions. The phenomena can also be linked to the possible advection of biomass burning byproducts from central Africa and crossing the gulf of Guinea resulting from the zonal transport (Menut et al., 2018; Flamant et al., 2018).
While sWA has received little attention regarding anthropogenic urban emissions, our study reports new observations and original analysis. Additional ground-truths and advanced satellite aerosol products or combination of products targeting on aerosol attribution are required to unravel the relative impact of anthropogenic versus natural aerosol emissions on atmospheric concentrations in this area of the world that is under a growing anthropogenic pressure.

*Data availability.* Handheld sun photometer and PM2.5 data are available at http://baobab.sedoo.fr/. AERONET sun photometer data are
430 available at https://aeronet.gsfc.nasa.gov/. MODIS aerosol data can be downloaded from https://ladsweb.modaps.eosdis.nasa.gov/.

*Acknowledgements.* The research leading to these results has received funding from the European Union 7th Framework Programme (FP7/2007-2013) under Grant Agreement no. 603502 (EU project DACCIWA: Dynamics-aerosol-chemistry-cloud interactions in West Africa). The authors greatly thank all the operators who contribute to the acquisition of handheld sun photometer observations at Lamto geophysical station, CEG Dantokpa and CEG Savè. We acknowledge the AERONET and PHOTONS sun-photometer networks, their staff
and the PI of the sites for their work to produce the dataset used in this study (http://aeronet.gsfc.nasa.gov/). Handheld CIMEL data were processed by I. Jankowiak (Laboratoire d'Optique Atmosphérique, Université Lille 1).

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
