# Peer review of "PM2.5 surface concentrations in southern West African urban areas based on sun photometer and satellite observations"

_Atmospheric Chemistry and Physics, 2020_

## Referee Comment (RC1) · Anonymous Referee #1 · 28 Aug 2020

General Comments This study analyzes aerosol properties and PM2.5 concentrations using sunphotometers (handheld and automatic-AERONET) and samples at selected sites (coastal and inland) in the south west Africa. This region is very interesting for aerosol studies due to seasonally changing meteorological conditions i.e, Harmattan north dry winds during winter and monsoon humid south winds in summer. This reverse atmospheric circulation, along with the influence of dust events and forest/agricultural fires result in contrasting aerosol types and properties. From this point of view, the current work may have its own importance, taking also into consideration the very good correlation between satellite AODs and PM2.5 concentrations over selected sites that allows the PM2.5 estimates at a long-term period and examination of the trends. All

the above issues, along with a rough classification of aerosol types are examined in this paper. However, the analysis, discussion, linkage of the present results with previous ones over the region and physical explanations of the aerosol properties related them with seasonality, meteorology and sources are missing or are rather poorly discussed throughout the manuscript. The analysis and discussions for each figure seem rather brief and do not emphasize on important issues of aerosol properties, mixing processes and connection of AOD with surface PM2.5 concentrations. In the following, I have specific comments for authors in a way to improve the scientific quality of the paper. In synopsis, detailed discussion of the results, enrichment of the literature and new analysis in some issues are necessary.

1) The methodology for the PM2.5 estimations from space is not described in detail. Authors should provide the analysis or even the scatter plots between MODIS-AODs and ground-based PM2.5 concentrations for each site and present the linear regressions and the converted factors.

2) The accuracy of the handheld sun-photometer measurements of the spectral AOD is very important, not only for the validation of the MODIS AODs and the PM estimations, but also for the extraction of intensive aerosol properties like the Angstrom exponent... So, these retrievals can be assured as of high accuracy. Authors may check it by applying a 2nd-order polynomial fit to the sun-photometer wavelengths (Eck et al., 2001; JGR). Since the only three wavelengths will give an excellent (R2=1) polynomial fit, the two constant terms A2, A1 may be used and compared with the Angstrom exponent at same wavelength band and in case of very low errors the A2-A1 should equal to Angstrom exponent (Schuster et al., 2006; JGR). You may see this application in Sharma et al. (2014, Aer.Air.Qual.Res), Tiwari et al. (2018, Env.Sci.Poll.Res.) and references therein. With such sensitivity test, you assure the accuracy of the manual sunphotometer retrievals, which may have perturbed due to invisible clouds or even due to not exact matching of the sun disc in the instrument's FOV.

3) Why did you use so long period (1 week) for filter samples? Usually, they are taken

on daily basis... Is this a technical reason or just to smooth the correlations with MODIS AODs, in order to avoid a larger daily variation? It is recommended to write something and discuss it more.

4) A critical point that has to be discussed regarding the measurement time series is the availability of sun photometer observations throughout the year and if there is an extent period (months, or season) with data missing. These large gaps may modify the AOD seasonality.

5) The present one constitutes a rather crude aerosol classification, as also recognized by the authors. It is based on three AE groups, while there are not thresholds at all for the AOD. Alternatively, it is recommended to use the classic AOD vs AE scatter plot for all the examined sites to identify major aerosol types, where the dominance of dust and urban pollution will be defined, especially if such a plot is applied for the dry and wet seasons. That plot would constitute a much better representation of the aerosol types and is highly recommended. There are numerous studies over the globe (some are cited in the manuscript) dealing with such analysis. Finally, the aerosol classification based on AOD vs AE scatter plot is a first rough classification able to discriminate between major aerosol types i.e. biomass burning, desert dust, sea salt, but not between absorbing urban aerosols from various combustion sources, i.e. such a classification is incapable of determining absorption aerosol properties (see Giles et al., 2012, JGR; Cazorla et al., 2013, ACP). Furthermore, at the end of the results, this classification is expanded on long-term periods using the MODIS AE values, which are not accurate enough for such aerosol classification studies and the results and trends may be subject of biases. All these should be clearly discussed in the manuscript.

6) Personally, I do not remember any other study that examined the PM2.5/AOD ratio and also there are no references for such studies here. However, I really doubt about the importance of this ratio, since it is strongly affected by the seasonally changing AOD and PM2.5 values. So, a similar ratio may define very contrasting aerosol regimes of low or high aerosol loading. Furthermore, I do not see a standard variation depending

on season or aerosol type, which may indicate specific characteristics of aerosols in a certain period. For example, in periods with high dust activity, I would expect a lower AOD ratio, since dust is mainly transported above. Also, I would expect higher values for the "urban-like" types, since urban pollution mainly confines within the boundary layer and not in the vertical, so the PM2.5 is usually increasing with higher rates than AOD. However, all these may be highly variable from site to site and here, the averaged values from all the sites mask the results. Furthermore, authors fail to discuss in detail the physical meaning of their results and/or the importance of them. In case they want to maintain this analysis, they should be more detailed in the discussion of the main results and what a low or high ratio value represents. In the current version, this analysis and discussions are not considered important for the paper.

Specific Comments Line 29: Add "in Chad" after depression.

Line 35: You may also see the recent global study about aerosol hot spot regions of dust, polluted-dust and smoke by Mehta et al. 2018. Mehta, M., Singh, N., Anshumali, 2018. Global trends of columnar and vertically distributed properties of aerosols with emphasis on dust, polluted dust and smoke - inferences from 10-year long CALIOP observations. Rem. Sens. Environ., 208, 120-132.

Line 36: You may refer some of the important studies that linked the ARF with the monsoon circulation and precipitation redistribution in the Arabian Sea and India, since such studies are rather rare or even absent in the south west-Africa.

Lines 36-39: These sentences can be merged.

Line 44: Especially for the issue of aerosol-type classification, authors may also see the global study by Hamill et al. (2016; Atmos. Envron.), which also covers the study region.

Lines 53-54: Another study (Sinha et al., 2015; Intern. J. Rem. Sens.) relates this AOD-PM regression with vertical profile of aerosols and meteorological parameters

(RH, Theta, wind) that strongly affect the PM vs. AOD correlation.

Line 63: Add "and" after time series.

Lines 91-92: Revise this sentence.

Lines 105-107: It is not clear if these are your results or other ones from previous validation studies... Please, clarify it. Also, a revision is needed in this sentence.

Line 143: "The overall range of AOD is (0.07, 3.8)." This does not make sense. In what station do you refer here? Also, the parenthesis confuses the reader that something else is missing here.

Lines 148-149: The rather significant variability in the mean AODs at Lamto site due to different periods of the measurements and number of observations necessitates discussing the availability of measurements throughout the year and if a specific season dominates (in number of measurements) against others...

Lines 156-157: Add ", respectively" at the end of the sentence.

Line 160: Revise as "... R=0.82, being R=0.90 between ..."

Lines 167-169: These characteristics should be discussed in view of aerosol sources, transport routes and dominant aerosol types in each season. Only presentation of the results without any explanation about their physical meaning is rather awkward.

Lines 170-172: Also, these results should be further discussed about aerosol types, sources and physical meaning. For example, why AE values are, on average, higher at the coastal stations despite the relative higher influence from sea-salt aerosols that are known to have low AEs? Furthermore, all sites throughout the year present rather low AEs, well below 1.0, except of some few cases, which has to be further commented by authors.

Line 178: Add "values" after RMSE.

Line 192-195: Some further discussions are needed here, regarding the suitability of using MODIS-derived AE values. According to my knowledge and several previous studies, MODIS-AE over land is highly biased since both Deep Blue and Dark Target algorithms used standard models and mixtures of them for the determination of the AE, which is not a measured but a computed parameter. So, the uncertainty increases significantly and this is also shown from the frequency distribution of the AE values from MODIS and sunphotometers. It would be nice, authors to present via graph, or even to give some values of comparison between MODIS-AE and sunphotometer AE in order to reveal the magnitude of the bias.

Line 197: AE does not depend on the aerosol optical properties. It is an intensive aerosol optical property by itself.

Line 209: Correct as "enables"

Line 210: Correct as "category".

Lines 215-216: This sentence needs revision in English grammar, syntax and typos errors.

Lines 217-218: For aerosol type classification in west Africa, authors may see the results by Hamill et al. (2016; Atm.Env.) about dominant aerosol types at several sites of the region.

Line 229: Correct as "ratios".

Line 235: Correct as "remain".

Lines 235-236: Which are the moderate PM2.5 and significant AOD values reported here? There is not a clear view of the levels of these values, which should be mentioned here.

Lines 236-238: This is only true for the "dust-like" aerosols and not for the other types. So, this is not a main findings of the analysis.

[Figure]

Lines 240-241: There is no analytic description of the satellite-derived PM2.5 concentrations. The methodology of these retrievals should be included in the manuscript. This analysis, based on weekly averages, is rather rough and not analytic about the association between dominant aerosol and PM2.5 levels. The scatter plot of AOD vs. AE may also include as third variable (color coded), the PM2.5 concentrations, so the reader would be able to see what type is associated with highest PM2.5. Statistics of such analysis may be also included in a Table. Authors may also provide the seasonality of the aerosol types and correlations between AOD and PM2.5 for each type separately.

Lines 247-251: The current excellent results regarding the PM2.5 estimates from satellite measurements should be compared with other studies over the globe, several of them cited in the Introduction section. In general, there is a lack of any comparison or even discussion of the present results with previous studies in the region. This constitutes a main drawback in the scientific presentation of the present study.

Line 254: It is not clear, since there is no discussion in the method, if authors used a unique PM2.5/AOD conversion fraction for the whole data or seasonal-dependent conversion factors, based on AOD vs PM2.5 scatter plots in each season. In other places over the globe, this seasonal dependence has been shown to be very important for the accuracy of the retrievals.

Lines 255-257: The critical here is to provide information about the PM2.5 levels above 25 $\mu$g m-3, which is the threshold for pollution established by EU. Authors should provide analysis about the seasonality of the PM2.5 exceedances and to associate them with the dusty period and the monsoon circulation.

Line 257: Correct as "Maximum is...".

Line 261: Explanation is needed for the much higher PM2.5 values in Cotonou, which also reflect higher AODs than those in Abidjan. Also, correct as "Abidjan".

[Figure]

Line 263: "PM2.5 concentrations are then multiplied by a factor of about 3 in 4 months.". This sentence is rather confusing without a clear meaning. It should be rephrased.

Figure 9: Why did you mix the observations from both sites? I think that this smooths the variability and possibly the trends. At any case, the trends should be examined separately for these sites. Also, authors are based on MODIS-AE values for the determination of the urban-like aerosol type. As discussed above, the MODIS-derived AEs are significantly biased and this should be mentioned in the text. However, in annual basis, the errors and biases are significantly reduced and trends may be examined but with caution. At any case, a full discussion about the biases occurred in such approaches should be given.

Line 265: Correct as "represents"

Lines 269 and 271. This is Figure 9.

Line 271: For such an explanation you may see the differences in MODIS AODs (likely significant lower values in 2018) or in the frequency distribution of the urban-like type. However, the annual PM2.5 and the trends in the two sites should be analyzed separately, so the current figures should be changed.

Lines 272-274: These results should be re-evaluated separately for the two sites. Also, here it is not referred that this increasing tendency corresponds to the urban-like aerosols. Also, there is no discussion about increasing trends in PM and anthropogenic pollution in the urban areas of the region, or even any comparison with previous works.

Conclusions section should be revised in view of the new results and discussions in the text.

---

## Referee Comment (RC2) · Anonymous Referee #2 · 6 Oct 2020

This paper combines satellite (MODIS) and ground-based (AERONET and hand-held) AOD with PM2.5 data to (1) evaluate MODIS AOD and (2) investigate the relationship between MODIS AOD and PM in southern West Africa. Empirical relationships between AOD and PM are created (based on season and an aerosol type proxy) and used to estimate trends in ground-level PM in the region.

The topic is in scope for the journal and the subject matter is important. The hand-held and PM data were presented by the authors in a previous paper in ACP (Djousse et al., 2018); this work is a natural extension of that (using the data collected and more in concert with MODIS) and contains sufficient novel material. The quality of language

is quite good, and the analysis, while fairly simple in parts, is explained well and uses statistics fairly appropriately (for example the authors acknowledge the skew in AOD distributions and treat this appropriately, while many authors do not).

There are some bits that are a bit unclear, I have a few thoughts on the data use, and I found some typographical issues. I recommend major revisions, mainly due to more detail needed in the AOD/PM ratio part. I would like to review the revised version. Comments are as follows:

1. As a general point about the MODIS retrievals: the authors use daily level 3 (L3) products (1 degree) from Aqua as the basis for both the comparison with Sun photometers and the PM prediction. For validation against Sun photometers it is more usual to use level 2 (L2) data with an averaging radius around 25 km to decrease discrepancies arising from real spatial and temporal variability. The authors might acknowledge that here. It is probably ok to use the L3 data if the goal is to make regional-scale PM analyses. But the purpose here (and the data collected) seems focused on the two cities. So I would suggest the authors might get L2 data and perform the same sun photometer comparison to see whether the same patterns hold. I suspect they might but without seeing the data we don't know. I acknowledge that this might not be feasible dependent on the computational resource available to the authors. I suppose doing both is a good way to test whether there is significant variability below the L3 scales.

2. The authors are also using (from their statements) an outdated version of the MODIS product: Collection 6 rather than 6.1. This was released several years ago now (in late 2017: https://atmosphere-imager.gsfc.nasa.gov/documentation/collection-61 ) so it is unfortunate to see Collection 6 still being used. Collection 6.1 has some algorithm updates as well as calibration updates which might affect the results. It would be preferable to repeat the analysis with the latest data version.

3. It would also be interesting to add a second satellite data set for an additional point of comparison. One option would be MODIS Terra, as the earlier overpass time

might mean different sampling due to cloud cover changes. Alternatively the authors might consider a different sensor or algorithm. There are many available during the 2014-2017 study period, but for the longer-term PM trend analysis the options are fewer. MISR has a narrower swath so there will be many fewer matchups with the Sun photometers, but its retrieval of aerosol properties has some more flexibility so it might perform better. OMI also has a nice smoke/dust aerosol type identification which might be useful here since part of the analysis involves relating AOD to PM based on aerosol type. So that could be a good addition. Another alternative is using a reanalysis product (e.g. MERRA2) which might also have surface concentration estimates. I am not saying this should be a requirement for publication, just something for the authors to consider.

4. Line 6: "Angstrom" should be written as "Ångström" here and throughout the paper. The paper is not always consistent. Line 110 and Table 1 have the ö but not the Å, for example, and Figure 3 has neither. This is not needed on line 119 though because there the authors are referring directly to the variable name.

5. Line 16: I think "S=" can be removed here.

6. Figure 1: again, not essential, but rather than have a greyscale map the authors might consider using e.g. a population map upon which to show the site locations.

7. Line 124: I would add parentheses around the EE expression as the interpretation of the +/- is ambiguous as written. I believe the correct representation is +/-(0.05+0.15xAOD) and not (+/- 0.05)+0.15xAOD.

8: Line 137: this should be IQR not IRQ.

9: Lines 144-145: the sentence says that the highest AOD was 3.8, but then says it was 3.7. This should be checked and corrected.

10. Lines 152-153: the offset in AE between two measurement types could well be related to calibration; the authors may wish to mention the study by Wagner and Silva

(2008) on this topic: https://acp.copernicus.org/articles/8/481/2008/

11. Line 156: AERONET collects data from dawn to dusk, while the hand-held instruments say they were used twice a day. Are these daily averages from all points or from the same times as the hand-held instruments?

12 Figure 5: my assumption is that the AE shown from MODIS Deep Blue and ocean algorithm here are for all points of the domain, and not only for the grid cells where there are sun photometer data. Is that correct? If so, some differences might also be expected due to real spatial variability in the AE at locations without sun photometer data. This should be mentioned.

13. Table 3: I am not sure I fully understand this as it took a few readings. It seems that the numbers not in brackets are the percentage of days from each category, using only the AE to split them. Then, the numbers in columns are the same, except considering only those days where the AOD was above the third quartile for that location, i.e. days where AOD was particularly high. So the table is contrasting the optical "type" of aerosols between sites, and also between the data set as a whole and those particular high-AOD days of concern for air quality purposes. Is that correct? I wonder if the numbers in parentheses should be given their own 3 columns with own subheader in the table. This is because it is the tendency as a reader to look at the number and the one next to it, and in this case they're not directly related, as the relevant comparison is between columns.

14. Figure 6, and associated discussion: this should be fleshed out more. First, is this figure showing mean and standard deviation of the ratio? This should be stated. However, another concern is whether mean and standard deviation are the right metrics to show; I suggest median and IQR could again be more appropriate. The authors do not show the raw data so there's no way to know whether there are e.g. outliers which are throwing off the mean ratios here. I suggest adding plots of the data (i.e. weekly AOD and PM for different seasons and type classifications). In theory, given constant

meteorology and composition, it is true that the ratio between AOD and PM should be a constant (the mass extinction efficiency and a factor based on height). However in practice factors such as changing composition, variations in aerosol vertical structure (e.g. whether or not the aerosols are mostly in the boundary layer), and moisture (as AOD is dependent on ambient RH while PM is not) will be important. See for example Sayer et al (2016) for the dependence of the ratio on some of these factors for smoke in Thailand: https://doi.org/10.4209/aaqr.2015.08.0500 The situation is a bit different here because the present study is weekly filter measurements (not continuous) but the general point remains. The authors need to show the data to provide justification that adopting a direct ratio approach, and reporting the mean and standard deviation (or changing to median and IQR), is an appropriate empirical parametrization here. It is not possible to judge from the material presented in the paper. As this figure is key to most of the rest of the paper, changes to this part of the analysis could affect the later results and discussion as well.

15. Figure 7 caption: "weakly" should be "weekly".

16. Figure 9 and associated discussion: given the high seasonality in AOD (and PM), as well as the potential for uneven satellite sampling through the year (due to e.g. cloud cover variations), I am not convinced that it makes sense to examine only annual trends in PM. I suggest adding seasonal analyses as well. This will provide more insight as to any changes. The authors might also add a plot showing e.g. the number of days with data in the average Jan, Feb etc from MODIS, so we can see whether sampling variations exist. A second point about trends is that rather than talk only about p values, it would be useful to mention the uncertainty estimate on the trend as well. Statistical significance and importance are not the same thing. For example, a statistically significant result might have a small magnitude which is not important for practical purposes. And a result that is not significant might be because either the estimated magnitude is small and the uncertainty is also fairly small (i.e. we can be confident there is not a large trend), or because we have a large uncertainty so can't

tell whether an effect is large or small (i.e. we can't be confident about the magnitude and/or sign of any trend). Reporting best estimate of trend and p value without the uncertainty estimate means we cannot directly tell which of these is the case here.

17. Data availability: the authors give Giovanni as the MODIS data source. This is mostly a visualization portal and may not have the latest/official data versions. I just checked there and they list collection 6.1 which makes it more surprising that the authors used the older collection 6. Note the main NASA search tool is https://earthdata.nasa.gov/ and the actual MODIS data portal for this product is LAADS, https://ladsweb.modaps.eosdis.nasa.gov/ .

---

## Author Comment (AC1) · 25 Nov 2020

General Comments

This study analyzes aerosol properties and PM2.5 concentrations using sunphotometers (handheld and automatic-AERONET) and samples at selected sites (coastal and inland) in the south west Africa. This region is very interesting for aerosol studies due to seasonally changing meteorological conditions i.e, Harmattan north dry winds during winter and monsoon humid south winds in summer. This reverse atmospheric circulation, along with the influence of dust events and forest/agricultural fires result in contrasting aerosol types and properties. From this point of view, the current work may have its own importance, taking also into consideration the very good correlation between satellite AODs and PM2.5 concentrations over selected sites that allows the PM2.5 estimates at a long-term period and examination of the trends. All the above issues, along with a rough classification of aerosol types are examined in this paper. However, the analysis, discussion, linkage of the present results with previous ones over the region and physical explanations of the aerosol properties related them with seasonality, meteorology and sources are missing or are rather poorly discussed throughout the manuscript. The analysis and discussions for each figure seem rather brief and do not emphasize on important issues of aerosol properties, mixing processes and connection of AOD with surface PM2.5 concentrations. In the following, I have specific comments for authors in a way to improve the scientific quality of the paper. In synopsis, detailed discussion of the results, enrichment of the literature and new analysis in some issues are necessary.

Dear Reviewer, Thank you for your time. We greatly appreciate your review and the references you have provided. The paper has been significantly modified and you will find below point-to-point answers to your review.

First, some general comments and information about the revised version for both reviewers.

- Firstly, we want to make clear about the region. We focus on the northern coastal part of the gulf of Guinea. Western Africa has a marked latitudinal gradient in ecosystems that largely impacts the emission and deposition of particules and trace gases (Adon et al., 2010). We define South Western Africa as delimited by the shore of gulf of Guinea and 9°N in agreement with previous authors (Kniffka et al., 2019). Most the previous studies on AOD observations were performed north of our region of interest, i.e. in the Sahel. The observations we report here are unprecedented in this area of the world so the comparison with previous studies is very limited.

- Reviewer #1 has raised a very important point on the misuse of MODIS AE over land to classify aerosols from AOD observations. So the methodology used for analysing the sun photometer data can't be directly transposed to MODIS record. Nonetheless similar conclusions can be drawn from the seasonal analysis.

- The methodology to estimate PM2.5 from AOD is based on a simple relationship of proportionality between both quantities that depends on the season and the aerosol type. The description of the method has been improved. Accurate estimate of the impact of the aerosol type on PM2.5/AOD ratio is limited by (i) the weekly frequency in the ground sample while aerosol type can change daily, (ii) the classification of aerosol type using Ångström exponent in a geographical area where mixing of aerosol type is high and pure characteristics are seldom observed. Notheless, the effect of the dust layer uplift on surface concentrations can be observed during the

intermediate period when the monsoon onset occurs.

1) The methodology for the PM2.5 estimations from space is not described in detail. Authors should provide the analysis or even the scatter plots between MODIS-AODs and ground-based PM2.5 concentrations for each site and present the linear regressions and the converted factors.

The methodology for PM2.5 estimation from AOD relies on a simple conversion factor that depends on the season and possibly on the Ångström exponent when this parameter is available. We provide equation 1 to explain the conversion. Conversion factors are presented in Figure 8 and retrieved PM2.5 are compared to observations for both sites in Figure 9. We discuss the error associated to the MODIS-derived PM2.5 in the text.

2) The accuracy of the handheld sun-photometer measurements of the spectral AOD is very important, not only for the validation of the MODIS AODs and the PM estimations,but also for the extraction of intensive aerosol properties like the Angstrom exponent...So, these retrievals can be assured as of high accuracy. Authors may check it by applying a 2nd-order polynomial fit to the sun-photometer wavelengths (Eck et al., 2001;JGR). Since the only three wavelengths will give an excellent (R2=1) polynomial fit,the two constant terms A2, A1 may be used and compared with the Angstrom exponent at same wavelength band and in case of very low errors the A2-A1 should equal to Angstrom exponent (Schuster et al., 2006; JGR). You may see this application in Sharma et al. (2014, Aer.Air.Qual.Res), Tiwari et al. (2018, Env.Sci.Poll.Res.) and references therein. With such sensitivity test, you assure the accuracy of the manual sunphotometer retrievals, which may have perturbed due to invisible clouds or even due to not exact matching of the sun disc in the instrument's FOV.

Thank you for the suggestion and the references. You suggest to check the AE value with regards to possible cloud contamination or error in collimation. A caveat should, however, be noted in your proposed method. We can't use a 2nd-order regression with only 3 points. The coefficients of the polynom are directly given by solving the linear equation system using the coordinates of the points. In the references you have given, the authors always use more wavelengths, 7 wavelenghts for Eck et al (2001) and 5 for Tiwari et al. (2018) and for a larger spectral range than in our study.

As AE is derived from the AOD-wavelenght log-log plot, it is subject to large error. The equation given by Hamonou et al. (JGR, 1999) shows that the absolute uncertainty in AE is sensitive to the spectral range and the relative uncertainty in AOD. Higher spectral range gives lower uncertainty on AE. However, as mentioned by Tiwari et al (2018), the departure from linearity leads to different AEs when considering different spectral ranges. In this paper, we use the same limited spectral range for the three different instrument. We estimate the AOD at 540 nm from AOD at 465 nm and AE then compare it to the measured AOD at 540 nm and should be within the measurement uncertainty, $\pm 0.02$. Any point showing an inconsistency in the wavelength dependence is filtered out. We have clarified this point in the text as well as added references to Sharma et al (2014) and Kaskaoutis and Kambezidis et al. (2008) on L97.

3) Why did you use so long period (1 week) for filter samples? Usually, they are taken

on daily basis... Is this a technical reason or just to smooth the correlations with MODIS AODs, in order to avoid a larger daily variation? It is recommended to write something and discuss it more.

For logistic reasons it was not possible to run 4 sites during 2.5 years on a daily basis. We use gravimetric measurements, not automatic microbalance. A weekly sampling period seems reasonable to catch the main temporal pattern of atmospheric aerosols over long period of time (Ouafo-Leumbe et al., 2017). However the comparison of our weekly samplings with daily sun photometer observations is not trivial, specifically when looking at intensive parameters like Ångström exponent. This has been clarified in the text L152.

4) A critical point that has to be discussed regarding the measurement time series is the availability of sun photometer observations throughout the year and if there is an extent period (months, or season) with data missing. These large gaps may modify the AOD seasonality.

All the sun photometer observations are already presented in Figure 2 and Figure 3 as daily values. As you can notice, there are some gaps in the observations as already mentioned in the text Line 142. We have discuss the seasonaly of sun photometer time series for the sites having a significant number of observations throughout the year. For the seasonal cycle presented in Figure 10, we have used the MODIS observations and no significant gap has been noticed. We define a new larger area (see text L336) that has a least one observations per day over the MODIS record.

5) The present one constitutes a rather crude aerosol classification, as also recognizedby the authors. It is based on three AE groups, while there are not thresholds at all for the AOD. Alternatively, it is recommended to use the classic AOD vs AE scatterplot for all the examined sites to identify major aerosol types, where the dominance of dust and urban pollution will be defined, especially if such a plot is applied for the dry and wet seasons. That plot would constitute a much better representation of the aerosol types and is highly recommended. There are numerous studies over the globe (some are cited in the manuscript) dealing with such analysis. Finally, the aerosol classification based on AOD vs AE scatter plot is a first rough classification able to discriminate between major aerosol types i.e. biomass burning, desert dust, sea salt,but not between absorbing urban aerosols from various combustion sources, i.e. such a classification is incapable of determining absorption aerosol properties (see Giles etal., 2012, JGR; Cazorla et al., 2013, ACP). F urthermore, at the end of the results, this classification is expanded on long-term periods using the MODIS AE values, which are not accurate enough for such aerosol classification studies and the results and trends may be subject of biases. All these should be clearly discussed in the manuscript.

Thank you for your remark. Correct. The AOD-versus-AE scatter plot is a well-known methodology to get a classification of the aerosol types, also the classification remains crude as well. Adding a threshold on AOD values is interesting to make a speciation between clean marine air (low AOD and low AE) and dust (high AOD and low AE), while for the separation between biomass burning events (high AOD and high AE) and urban pollution events (high AOD and high AE) is rather difficult since, as you say in your

comment, this classification with AOD can't be used to determine aerosol absorption. In our dataset, the category 'clean marine air' is statistically unsignificant because of mixing with polluted air and continental aerosols. Moreover, there is a risk of over interpretation of the AE results for weak AOD because of uncertainty in AE depends on the AOD level (eg. Hamonou et al. 1999). However, we have followed your remark and we present the AOD vs AE plots for the differents sites and seasons. We also define a threshold on AOD to better identified dust event compared to mixing cases.

You comment on the MODIS AE has rised an important point. We have further investigated the use of MODIS AE (DB, OCEAN and Land algorithm) and finally end to the same conclusions as yours. At the coastal sites, both AE from the DT and OCEAN algorithm tends to reproduce the seasonal variability that is observed by the sun photometer. However there is no correlation between MODIS AE and sun photometer AE and MODIS AE can't be used to classify the daily observations.

*Please refer to L250 and the new figure 6*

6) Personally, I do not remember any other study that examined the PM2.5/AOD ratio and also there are no references for such studies here. However, I really doubt about the importance of this ratio, since it is strongly affected by the seasonally changing AOD and PM2.5 values. So, a similar ratio may define very contrasting aerosol regimes of low or high aerosol loading. Furthermore, I do not see a standard variation depending on season or aerosol type, which may indicate specific characteristics of aerosols in a certain period. For example, in periods with high dust activity, I would expect a lower AOD ratio, since dust is mainly transported above. Also, I would expect higher values for the "urban-like" types, since urban pollution mainly confines within the boundary layer and not in the vertical, so the PM2.5 is usually increasing with higher rates than AOD. However, all these may be highly variable from site to site and here, the averaged values from all the sites mask the results. Furthermore, authors fail to discuss in detail the physical meaning of their results and/or the importance of them. In case they want to maintain this analysis, they should be more detailed in the discussion of the main results and what a low or high ratio value represents. In the current version, this analysis and discussions are not considered important for the paper.

As mentionned in the introduction, there is an abundant litterature on AOD to PM conversion and a large diversity in the proposed methods albeit no consensus. The PM2.5/AOD is basically the amount of PM2.5 that is expected per unit of AOD. It was first promoted by van Donkelaar et al. (2010) as a conversion factor (Zheng et al., 2017; Yang et al., 2019). You are right, a similar ratio may define very constrasted aerosol regimes, including different aerosol types or vertical profiles. However we know that the most important phenomum that impact the AOD to PM2.5 relationship in this area is the transport of mineral dust from the Sahel and the shift of the dusty layer toward upper altitude during the pre-monsoon period. So we have analyzed the PM2.5/AOD ratio as a function of the season and aerosol type (Figure 8).

Standard errors on PM2.5/AOD ratios are already reported in Figure 6. We are sorry to say that your analysis on the PM2.5/AOD ratio is not correct. During winter dry period the dust is advected near the ground by the Harmattan wind so the PM2.5/AOD ratio is high (about 50 $\mu g/m^3$ per unit of AOD).

During the transition period (April-June), the dust layer is uplifted by the monsoon flow. Although the AOD is high, the PM2.5 remains low, leading to a PM2.5 ratio for dusty days of about half the one for the winter period. This phenomemum is similar for the

whole area and we pooled the dataset to increase statistical significance of the results. However, as suggested in your review, we provide the comparison between observed and derived PM2.5 for each site separately.

*Text has been changed on L280. Figure 9 has been changed to show both sites.*

Specific Comments

Line 29: Add "in Chad" after depression.

Done.

Line 35: You may also see the recent global study about aerosol hot spot regions of dust, polluted-dust and smoke by Mehta et al. 2018. Mehta, M., Singh, N., Anshumali,2018. Global trends of columnar and vertically distributed properties of aerosols with emphasis on dust, polluted dust and smoke - inferences from 10-year long CALIOP observations. Rem. Sens. Environ., 208, 120-132.

Thank you for providing usefull references. This reference has been added Line 35.

Line 36: You may refer some of the important studies that linked the ARF with the monsoon circulation and precipitation redistribution in the Arabian Sea and India, since such studies are rather rare or even absent in the south west-Africa.

Thank you for the suggestion. Actually our sentence was awkward.

*Please see new paragraph L36 in the revised document.*

Lines 36-39: These sentences can be merged.

Done.

Line 44: Especially for the issue of aerosol-type classification, authors may also see the global study by Hamill et al. (2016; Atmos. Envron.), which also covers the study region.

Thank you for the reference. We have added Hamill et al. (2016) to the list of references Line 44.

Lines 53-54: Another study (Sinha et al., 2015; Intern. J. Rem. Sens.) relates this AOD-PM regression with vertical profile of aerosols and meteorological parameters (RH, Theta, wind) that strongly affect the PM vs. AOD correlation.

This new reference has been added Line 63.

Line 63: Add "and" after time series.

Done.

Lines 91-92: Revise this sentence.

Done.

Lines 105-107: It is not clear if these are your results or other ones from previous validation studies... Please, clarify it. Also, a revision is needed in this sentence.

CALITOO handheld sunphotometer were calibrated using Langley plot at high altitude Izana observatory. The calibration was check using coincident AERONET observations before and after field experiment We have modified the text as follows.

*CALITOO observations were compared to coincident AERONET observations before and after the field experiment. The total uncertainty in AOD is estimated to $\pm 0.02$ for all the wavelengths. Post-field measurements indicates a change of about 1% per year in the calibration.*

Line 143: "The overall range of AOD is (0.07, 3.8)." This does not make sense. In what station do you refer here? Also, the parenthesis confuses the reader that something else is missing here.

The overall range means considering all the stations together. Brackets indicates (minimum value, maximum value). As it appears not clear for the reader, we have changed this sentence. See L170.

Lines 148-149: The rather significant variability in the mean AODs at Lamto site due to different periods of the measurements and number of observations necessitates discussing the availability of measurements throughout the year and if a specific season dominates (in number of measurements) against others...

The timeseries at Lamto are presented on Fig. 2. No season dominates for HCC or CALITOO although the observations corresponds to different years. AERONET dataset is rather limited (see table 1, revised as there were an inconsistency in the ending date of the measurements). We have recall the observation period for Lamto in the main text L176.

Lines 156-157: Add ", respectively" at the end of the sentence.

Done.

Line 160: Revise as "... R=0.82, being R=0.90 between ..."

Done.

Lines 167-169: These characteristics should be discussed in view of aerosol sources,transport routes and dominant aerosol types in each season. Only presentation of the results without any explanation about their physical meaning is rather awkward.
Lines 170-172: Also, these results should be further discussed about aerosol types,sources and physical meaning. For example, why AE values are, on average, higher at the coastal stations despite the relative higher influence from sea-salt aerosols that are known to have low AEs? Furthermore, all sites throughout the year present rather low AEs, well below 1.0, except of some few cases, which has to be further commented by authors.

This section describes the observations while the physical meaning is discussed later in the text. However we have followed your recommandations and give the explanation of the physical meaning in this paragraph.Coastal sites are located in large conurbations so the AE remains higher than at the inland sites. AE values are further discussed in the classification section. See L199 and L205.

Line 178: Add "values" after RMSE.

Done.

Line 192-195: Some further discussions are needed here, regarding the suitability of using MODIS-derived AE values. According to my knowledge and several previous studies, MODIS-AE over land is highly biased since both Deep Blue and Dark Target algorithms used standard models and mixtures of them for the determination of the AE, which is not a measured but a computed parameter. So, the uncertainty increases significantly and this is also shown from the frequency distribution of the AE values from MODIS and sunphotometers. It would be nice, authors to present via graph, or even to give some values of comparison between MODIS-AE and sunphotometer AE in order to reveal the magnitude of the bias.

Correct. See your preceding comment new Figure 6 and corresponding text.

Line 197: AE does not depend on the aerosol optical properties. It is an intensive aerosol optical property by itself.

Correct but following the definition given by Ångström, AE depends on the aerosol extinction coefficients, which is an aerosol optical properties. Sentence has been revised as follows.

*AE is an intensive aerosol optical parameter and depends on the spectral aerosol extinction coefficient (Nakajima et al., 1996; Eck et al., 1999; Holben et al., 2001). AE is influenced by the aerosol size distribution and is commonly used to identify aerosol types (Léon et al., 1999; Kaskaoutis et al., 2009; Perrone et al., 2005).*

Line 209: Correct as "enables"

Done.

Line 210: Correct as "category".

Done.

Lines 215-216: This sentence needs revision in English grammar, syntax and typos errors.

Done.

Lines 217-218: For aerosol type classification in west Africa, authors may see the results by Hamill et al. (2016; Atm.Env.) about dominant aerosol types at several sites of the region.

Thank you for the reference. Actually Hamill et al (2016) is out of the region. So no comparison can directly apply. However we have used this reference for dust source activity north of our domain. We now cite this paper L271.

Line 229: Correct as "ratios".

Done.

Line 235: Correct as "remain".

Done.

Lines 235-236: Which are the moderate PM2.5 and significant AOD values reported here? There is not a clear view of the levels of these values, which should be mentioned here.

We now give the corresponding values. Values are also reported in Figure 9.

Lines 236-238: This is only true for the "dust-like" aerosols and not for the other types. So, this is not a main findings of the analysis. We don't get your point here.

Lines 240-241: There is no analytic description of the satellite-derived PM2.5 concentrations. The methodology of these retrievals should be included in the manuscript. This analysis, based on weekly averages, is rather rough and not analytic about the association between dominant aerosol and PM2.5 levels. The scatter plot of AOD vs. AE may also

include as third variable (color coded), the PM2.5 concentrations, so the reader would be able to see what type is associated with highest PM2.5. Statistics of such analysis may be also included in a Table. Authors may also provide the seasonality of the aerosol types and correlations between AOD and PM2.5 for each type separately.

The text associated to Figure 7 (now Figure 9) has been improved. Several points were unclear. The color code on Figure 9 directly give which type is associated to PM2.5 levels. In the first version we have mixed MODIS observations and sun photometer observations to increase the number of comparison points (specifically for Abidjan where we have less sun photometer observations). However as in this revised version we don't rely on MODIS AE, we only use sun photometer observations in Figure 9.

Lines 247-251: The current excellent results regarding the PM2.5 estimates from satellite measurements should be compared with other studies over the globe, several of them cited in the Introduction section. In general, there is a lack of any comparison or even discussion of the present results with previous studies in the region. This constitutes a main drawback in the scientific presentation of the present study.

We agree on this point however it is not a main drawback. Not all the authors report the same metrics and there is no scientific agreement on which algoritm should be used to derive PM2.5. A general discussion on RMSE for the different type of algoritm is out of the scope of this paper. Moreover there is no similar study in the area targeted in our paper, so the comparison remains limited. Nonetheless we propose to mention L336 that the retrieved RMSE are within the range of previous studies and we cite Sinha et al. (2015) and Ma et al. (2015).

Line 254: It is not clear, since there is no discussion in the method, if authors used a unique PM2.5/AOD conversion fraction for the whole data or seasonal-dependent conversion factors, based on AOD vs PM2.5 scatter plots in each season. In other places over the globe, this seasonal dependence has been shown to be very important for the accuracy of the retrievals.

Correct. Now it is explicitly written that we use the seasonaly adjusted ratios presented in Figure 9.

Lines 255-257: The critical here is to provide information about the PM2.5 levels above $25\mu gm^{-3}$, which is the threshold for pollution established by EU. Authors should provide analysis about the seasonality of the PM2.5 exceedances and to associate them with the dusty period and the monsoon circulation.

The EU target of $25\mu gm^{-3}$ is for the annual mean. We report the years being below this threshold. We also report daily exceedances above $35\mu gm^{-3}$.

Line 257: Correct as "Maximum is...".

Done.

Line 261: Explanation is needed for the much higher PM2.5 values in Cotonou, which also reflect higher AODs than those in Abidjan. Also, correct as "Abidjan".

The data used for figure 8 as well as the figure have been revised. We still observed a large difference in April (23% higher in Cotonou) that is clearly attributed to a change in the contribution of coarse dust(+30% in Cotonou) while the contribution of other types remains the same. So our hypothesis is a higher contribution of dust during the dry and intermediate period over Cotonou. Further validation of this hypothesis will need both numerical modelling of the dust transport and in situ chemical speciation.

*See text L365 and new figure 10.*

Line 263: "PM2.5 concentrations are then multiplied by a factor of about 3 in 4 months.". This sentence is rather confusing without a clear meaning. It should be rephrased.

The sentence described the increase in PM2.5 concentrations between September and December. Now rephrased

*see revised text L371*

Figure 9: Why did you mix the observations from both sites? I think that this smooths the variability and possibly the trends. At any case, the trends should be examined separately for these sites. Also, authors are based on MODIS-AE values for the determination of the urban-like aerosol type. As discussed above, the MODIS-derived AEs are significantly biased and this should be mentioned in the text. However, in annual basis, the errors and biases are significantly reduced and trends may be examined but with caution. At any case, a full discussion about the biases occurred in such approaches should be given.

In the first version we have mixed the observations from both sites to increase the statistical significance of the results. We now introduce a new wider geographical area called SWA that includes both cities but provides more observations. Figure 9 have been changed to show (i) the monthly mean MODIS-derived PM2.5 (ii) the trend in the short dry period. MODIS AE is not used any more in the long-term analysis (see your preceeding remark). We report the 95% confidence interval for the trend indicating that the trend remains uncertain. As PM2.5 is proportional to AOD, any bias occuring in AOD, including calibration drift, will affect the PM2.5 concentrations.

*see revised text L385 and new reference to Sayer et al. 2019*

Line 265: Correct as "represents"Lines 269 and 271. This is Figure 9.

Done.

Line 271: For such an explanation you may see the differences in MODIS AODs (likely significant lower values in 2018) or in the frequency distribution of the urban-like type. However, the annual PM2.5 and the trends in the two sites should be analyzed separately, so the current figures should be changed.

Yes. AODs for 2018 (and 2019 added now) are lower than the previous years. We observed a 20% decrease in AOD annual average between 2017 and 2019. As stated earlier, the change in the frequency of urban-like can't be addressed from MODIS. However we now plot the monthly mean PM2.5 (SWA area in Figure 9) that best highlights the decrease in the recent years what ever is the season. The differences in monthly means between Abdijan and Cotonou and the SWA are not significant (see also the monthly annual cycle in Figure 10) for the trend analysis and not displayed on Figure 11.

*see revised text L376 and Figure 11.*

Lines 272-274: These results should be re-evaluated separately for the two sites. Also, here it is not referred that this increasing tendency corresponds to the urban-like aerosols. Also, there is no discussion about increasing trends in PM and anthropogenic pollution in the urban areas of the region, or even any comparison with previous works.

Due to the large uncertainty in the trend analysis, discussing the difference between the two sites is not relevant. We choose to present the trend for a broader area that encompassed both cities have at least one observations per day so any bias due to sampling is avoided. There no such previous work over the area so comparison remains very limited and moreover regional scale emission scenario for this particular area are scarce. As stated in the conclusion, the trend is consistent with existing emission scenarii.

*Please refer to the conclusion section.*

Conclusions section should be revised in view of the new results and discussions in the text. Conclusions and abstract have been revised accordingly.

**References**

[revised manuscript text omitted]

---

## Author Comment (AC2) · 25 Nov 2020

This paper combines satellite (MODIS) and ground-based (AERONET and hand-held) AOD with PM2.5 data to (1) evaluate MODIS AOD and (2) investigate the relationship between MODIS AOD and PM in southern West Africa. Empirical relationships between AOD and PM are created (based on season and an aerosol type proxy) and used to estimate trends in ground-level PM in the region. The topic is in scope for the journal and the subject matter is important. The hand-held and PM data were presented by the authors in a previous paper in ACP (Djousse etal., 2018); this work is a natural extension of that (using the data collected and morein concert with MODIS) and contains sufficient novel material. The quality of language is quite good, and the analysis, while fairly simple in parts, is explained well and uses statistics fairly appropriately (for example the authors acknowledge the skew in AOD distributions and treat this appropriately, while many authors do not). There are some bits that are a bit unclear, I have a few thoughts on the data use, and I found some typographical issues. I recommend major revisions, mainly due to more detail needed in the AOD/PM ratio part. I would like to review the revised version.
Comments are as follows:

Dear Reviewer,
Thank you for your time. We greatly appreciate your review and the references you have provided. The paper has been significantly modified and you will find below point-to-point answers to your review.

First, some general comments and information about the revised version for both reviewers.

- Firstly, we want to make clear about the region. We focus on the northern coastal part of the gulf of Guinea. Western Africa has a marked latitudinal gradient in ecosystems that largely impacts the emission and deposition of particules and trace gases (Adon et al., 2010). We define South Western Africa as delimited by the shore of gulf of Guinea and 9°N in agreement with previous authors (Kniffka et al., 2019). Most the previous studies on AOD observations were performed north of our region of interest, i.e. in the Sahel. The observations we report here are unprecedented in this area of the world so the comparison with previous studies is very limited.

- Reviewer #1 has raised a very important point on the misuse of MODIS AE over land to classify aerosols from AOD observations. So the methodology used for analysing the sun photometer data can't be directly transposed to MODIS record. Nonetheless similar conclusions can be drawn from the seasonal analysis.

- The methodology to estimate PM2.5 from AOD is based on a simple relationship of proportionality between both quantities that depends on the season and the aerosol type. The description of the method has been improved. Accurate estimate of

the impact of the aerosol type on PM2.5/AOD ratio is limited by (i) the weekly frequency in the ground sample while aerosol type can change daily, (ii) the classification of aerosol type using Ångström exponent in a geographical area where mixing of aerosol type is high and pure characteristics are seldom observed. Notheless, the effect of the dust layer uplift on surface concentrations can be observed during the intermediate period when the monsoon onset occurs.

1. As a general point about the MODIS retrievals: the authors use daily level 3 (L3) products (1 degree) from Aqua as the basis for both the comparison with Sun photometers and the PM prediction. For validation against Sun photometers it is more usual to use level 2 (L2) data with an averaging radius around 25 km to decrease discrepancies arising from real spatial and temporal variability. The authors might acknowledge that here. It is probably ok to use the L3 data if the goal is to make regional-scale PM analyses. But the purpose here (and the data collected) seems focused on the two cities. So I would suggest the authors might get L2 data and perform the same sunphotometer comparison to see whether the same patterns hold. I suspect they might but without seeing the data we don't know. I acknowledge that this might not be feasible dependent on the computational resource available to the authors. I suppose doing both is a good way to test whether there is significant variability below the L3 scales.

We agree with the reviewer that validation exercises are usually applied to the MODIS level 2 pixels rather than gridded products. However our objective is not to valide the retrieval algorithms but to address the ability of MODIS data to reflect aerosol changes in a specific area. Doing the validation of both L2 and L3 products will lead to further investigations in the statistical representativity of L3 versus L2 that is not the purpose of the paper.

*Please refer to L125 in the revised version.*

2. The authors are also using (from their statements) an outdated version of the MODIS product: Collection 6 rather than 6.1. This was released several years ago now (in late 2017: `https://atmosphere-imager.gsfc.nasa.gov/documentation/collection-61` ) so it is unfortunate to see Collection 6 still being used. Collection 6.1 has some algorithm updates as well as calibration updates which might affect the results. It would be preferable to repeat the analysis with the latest data version.

You have rised an important point. Our archive was not uptodate and actually a mixed between Version 6.0 and 6.1. This problem has been fixed although with there is no large impact on the results.

3. It would also be interesting to add a second satellite data set for an additional point of comparison. One option would be MODIS Terra, as the earlier overpass time might mean different sampling due to cloud cover changes. Alternatively the authors might consider a different sensor or algorithm. There are many available during the 2014-2017 study

period, but for the longer-term PM trend analysis the options are fewer. MISR has a narrower swath so there will be many fewer matchups with the Sun photometers, but its retrieval of aerosol properties has some more flexibility so it might perform better. OMI also has a nice smoke/dust aerosol type identification which might be useful here since part of the analysis involves relating AOD to PM based on aerosol type. So that could be a good addition. Another alternative is using a reanalysis product (e.g. MERRA2) which might also have surface concentration estimates. I am not saying this should be a requirement for publication, just something for the authors to consider.

*Thank you for your suggestion. We agree that a mixed of different satellite products along with reanalysis like MERRA2 could provide interesting information on AOD-PM relationship in our area of interest. However there are not so much options for long-term analysis as you have noticed. We have selected the MODIS record as it is comprehensive and well referenced. CALIPSO would be also an interesting alternative that requires further investigations. We have followed Wei et al. (2019) recommandations of using AQUA products.*

*Please refer to L127 of the revised document.*

4. Line 6: "Angstrom" should be written as "Ångström" here and throughout the paper. The paper is not always consistent. Line 110 and Table 1 have the ö but not the Å, for example, and Figure 3 has neither. This is not needed on line 119 though because there the authors are referring directly to the variable name.

*We now use the correct writing of Ångström in the text, bibliography, and in the figure captions. AE is used for figure labels.*

5. Line 16: I think "S=" can be removed here.

*Done.*

6. Figure 1: again, not essential, but rather than have a greyscale map the authors might consider using e.g. a population map upon which to show the site locations.

*Figure 1 has been updated with the geographical location and population of cities having more than 1,000 inhab.*

7. Line 124: I would add parentheses around the EE expression as the interpretation of the +/- is ambiguous as written. I believe the correct representation is +/-(0.05+0.15xAOD) and not (+/- 0.05)+0.15xAOD.

Done.

8: Line 137: this should be IQR not IRQ.

Done.

9: Lines 144-145: the sentence says that the highest AOD was 3.8, but then says it was 3.7. This should be checked and corrected.

Correct value is 3.76. The text has been modified.

*See revised text L170*

10. Lines 152-153: the offset in AE between two measurement types could well be related to calibration; the authors may wish to mention the study by Wagner and Silva (2008) on this topic: https://acp.copernicus.org/articles/8/481/2008/

Thank you for the reference. Very interesting. Rather than related to calibration, the difference might be due to the difference in the AOD statistical distribution. Indeed the observations were not acquired during the same period and the AOD distributions are not the same. We have added this possible explanation in the text with the corresponding reference. Moreover, we have better discussed the comparison on coincident observations between AERONET and CALITOO in Lamto. Despite a short sample period, the agreement is excellent (see added figure). Note there is an error in the sampling period in Table 1 but Figure 2 and 3 shows the overlap period.

*See revised text L181.*

11. Line 156: AERONET collects data from dawn to dusk, while the hand-held instruments say they were used twice a day. Are these daily averages from all points or from the same times as the hand-held instruments?

AERONET observations are daily average. Now mentionned in the text L84.

12 Figure 5: my assumption is that the AE shown from MODIS Deep Blue and ocean algorithm here are for all points of the domain, and not only for the grid cells wherethere are sun photometer data. Is that correct? If so, some differences might also be expected due to real spatial variability in the AE at locations without sun photometer data. This should be mentioned.

The comparison of MODIS AE values have been revised to better show the difference at both the coastal and inland sites. We used only the measurements corresponding to the grid cell where the sun photometers are located.

*Please refer to revised Figure 6 and text L215.*

13. Table 3: I am not sure I fully understand this as it took a few readings. It seems that the numbers not in brackets are the percentage of days from each category, using only the AE to split them. Then, the numbers in columns are the same, except considering only those days where the AOD was above the third quartile for that location, i.e.days where AOD was particularly high. So the table is contrasting the optical "type" ofaerosols between sites, and also between the data set as a whole and those particularhigh-AOD days of concern for air quality purposes. Is that correct? I wonder if thenumbers in parentheses should be given their own 3 columns with own subheader inthe table. This is because it is the tendency as a reader to look at the number and the one next to it, and in this case they're not directly related, as the relevant comparison is between columns.

We agree that the information provided in the Table 3 were not easy to read. We have simplified this table and now provide only the percentage of daily observations for each category and each site. Please note that the 'Dust' category has been updated by using a threshold on AOD.

*See new table 3.*

14. Figure 6, and associated discussion: this should be fleshed out more. First, is this figure showing mean and standard deviation of the ratio? This should be stated. However, another concern is whether mean and standard deviation are the right metrics to show; I suggest median and IQR could again be more appropriate. The authors do not show the raw data so there's no way to know whether there are e.g. outliers which are throwing off the mean ratios here. I suggest adding plots of the data (i.e. weekly AOD and PM for different seasons and type classifications). In theory, given constant meteorology and composition, it is true that the ratio between AOD and PM should be a constant (the mass extinction efficiency and a factor based on height). However in practice factors such as changing composition, variations in aerosol vertical structure (e.g. whether or not the aerosols are mostly in the boundary layer), and moisture (as AOD is dependent on ambient RH while PM is not) will be important. See for example Sayer et al (2016) for the dependence of the ratio on some of these factors for smoke in Thailand: https://doi.org/10.4209/aaqr.2015.08.0500 The situation is a bit different here because the present study is weekly filter measurements (not continuous) but the general point remains. The authors need to show the data to provide justification that adopting a direct ratio approach, and reporting the mean and standard deviation (or changing to median and IQR), is an appropriate empirical parametrization here. It is not possible to judge from the material presented in the paper. As this figure is key to most of the rest of the paper, changes to this part of the analysis could affect the later results and discussion as well.

We agree with your remark that the actual PM2.5/AOD ratio is subject to a large variability due to the modification of meteorological conditions or changes in aerosol chemical composition. We have added your proposed reference (Sayer et al., 2016) in the introduction section where we discuss previous studies. So far there is no scientific agreement on a universal method to convert aerosol optical parameters into surface concentrations however it is generally admitted that AOD and PM2.5 are correlated. Conversely, for steady surface concentrations and columnar AODs, a poor correlation between both quantities doesn't mean that a PM2.5/AOD ratio is not appropriate. Errors on the PM2.5/AOD ratio can then be estimated from the standard deviations of both quantities. The time series of coincident weekly observations already reported in Figure 9 (now for each site) provide posterior justification that the PM2.5 is proportional to AOD by period of time and that the aerosol type may also has some influence on the proportionality. So the coefficient are estimated by season and by aerosol type. The standard error reported on Figure 6 reflects that any change in AOD is not associated to an exact proportional change in PM2.5 over time. However and as you mentionned, there is an additionnal difficulty due to the different sampling period (weekly PM2.5 versus daily AOD). We handle this difficulty by estimating the relative influence of each aerosol type during a week.

*Please see additional information in the method section L298 and Figure 9.*

15. Figure 7 caption: "weakly" should be "weekly".

Done.

16. Figure 9 and associated discussion: given the high seasonality in AOD (and PM),as well as the potential for uneven satellite sampling through the year (due to e.g. cloud cover variations), I am not convinced that it makes sense to examine only annual trends in PM. I suggest adding seasonal analyses as well. This will provide more insight as to any changes. The authors might also add a plot showing e.g. the number of days with data in the average Jan, Feb etc from MODIS, so we can see whether sampling variations exist. A second point about trends is that rather than talk only about p values, it would be useful to mention the uncertainty estimate on the trend as well. Statistical significance and importance are not the same thing. For example, a statistically significant result might have a small magnitude which is not important for practical purposes. And a result that is not significant might be because either the estimated magnitude is small and the uncertainty is also fairly small (i.e. we can be confident there is not a large trend), or because we have a large uncertainty so can't tell whether an effect is large or small (i.e. we can't be confident about the magnitude and/or sign of any trend). Reporting best estimate of trend and p value without the uncertainty estimate means we cannot directly tell which of these is the case here.

Correct. The trend analysis has been totaly revised. Following your recommandation, we now provide the monthly mean concentrations and the seasonaly adjusted trend. We have reduced the potential impact of uneven sampling by selecting a broader area along the shore line. This larger area has at least one observation per day. Yes we totally agree with your remark however the uncertainty on the Thiel-Sen's slope were already reported in original text L273. The monotonic increasing trend is significant however the exact magnitude remains highly uncertain.

17. Data availability: the authors give Giovanni as the MODIS data source. This is mostly a visualization portal and may not have the latest/official data versions. I just checked there and they list collection 6.1 which makes it more surprising that the authors used the older collection 6. Note the main NASA search tool is `https://earthdata.nasa.gov/` and the actual MODIS data portal for this product is LAADS, `https://ladsweb.modaps.eosdis.nasa.gov/`.

Correct. Actually we did a first attempt of using MODIS L3 products from the Giovanni portal that also provides an extraction tool. Then we used the a mirror archive but the archive wasn't uptodate. Now we use the data from the LAADS portal `https://ladsweb.modaps.eosdis.nasa.gov/`.

*Thank you for your warning on the MODIS Version. We have changed the data availability section.*

**References**

Adon, M., Galy-Lacaux, C., Yoboué, V., Delon, C., Lacaux, J. P., Castera, P., Gardrat, E., Pienaar, J., Al Ourabi, H., Laouali, D., Diop, B., Sigha-Nkamdjou, L., Akpo, A., Tathy, J. P., Lavenu, F., and Mougin, E.: Long Term Measurements of Sulfur Dioxide, Nitrogen Dioxide, Ammonia, Nitric Acid and Ozone in Africa Using Passive Samplers, Atmos. Chem. Phys., 10, 7467–7487, doi:10.5194/acp-10-7467-2010, 2010.

Kniffka, A., Knippertz, P., and Fink, A. H.: The Role of Low-Level Clouds in the West African Monsoon System, Atmospheric Chemistry and Physics, 19, 1623–1647, doi:10.5194/acp-19-1623-2019, 2019.

Sayer, A. M., Hsu, N. C., Hsiao, T.-C., Pantina, P., Kuo, F., Ou-Yang, C.-F., Tsay, S.-C., Holben, B. N., Janjai, S., Chantara, S., Wang, S.-H., Loftus, A. M., and Lin, N.-H.: In-Situ and Remotely-Sensed Observations of Biomass Burning Aerosols at Doi Ang Khang, Thailand during 7-SEAS/BASELInE 2015, Aerosol and Air Quality Research, 16, 2786–2801, doi:10.4209/aaqr.2015.08.0500, 2016.

Wei, J., Peng, Y., Guo, J., and Sun, L.: Performance of MODIS Collection 6.1 Level 3 Aerosol Products in Spatial-Temporal Variations over Land, Atmospheric Environment, 206, 30–44, doi:10.1016/j.atmosenv.2019.03.001, 2019.